# Recommendations with Sparse Comparison Data: Provably Fast Convergence for Nonconvex Matrix Factorization

**Suryanarayana Sankagiri** [1]   **Jalal Etesami** [2]   **Matthias Grossglauser** [1]

## Abstract

In this paper, we consider a recommender system that elicits user feedback through pairwise comparisons instead of ratings. We study the problem of learning personalised preferences from such comparison data via collaborative filtering. Similar to the classical matrix completion setting, we assume that users and items are endowed with low-dimensional latent features. These features give rise to user-item utilities, and the comparison outcomes are governed by a discrete choice model over these utilities. The task of learning these features is then formulated as a maximum likelihood problem over the comparison dataset. Despite the resulting optimization problem being nonconvex, we show that gradient-based methods converge exponentially to the latent features, given a warm start. Importantly, this result holds in a sparse data regime, where each user compares only a few pairs of items. Our main technical contribution is to extend key concentration results commonly used in matrix completion to our model. Simulations reveal that the empirical performance of the method exceeds theoretical predictions, even when some assumptions are relaxed. Our work demonstrates that learning personalised recommendations from comparison data is both computationally and statistically efficient.

## 1. Introduction

Recommender systems are central to modern streaming platforms and digital marketplaces, where they curate personalized selections for each user from the vast set of items these platforms host. Algorithms powering such systems learn users' preferences based on the feedback that they provide, *e.g.*, the ratings on the items they have consumed. A classical method of learning from ratings data is *matrix completion*, which is based on the following modeling assumptions. Each user and item is endowed with a low-dimensional feature vector, and their inner product is taken to be the user-item utility. The ratings are assumed to be noisy reflections of this utility. By fitting this model to the available rating data, the system can learn to predict users' preferences over unseen items as well. This approach has been immensely successful in practice (Koren et al., 2009) and is also supported by strong theoretical foundations (Ge et al., 2016).

This work focuses on recommender systems that learn from pairwise preference comparisons instead of ratings. One reason to consider such a setting is that comparison data are widely available as implicit feedback—for instance, when a user clicks on one of four options, it suggests a preference for the selected item over the others. Additionally, we believe explicitly collecting comparison feedback instead of ratings can be beneficial for several reasons: (i) comparisons naturally cancel out user biases in ratings (Shah et al., 2013); (ii) they avoid the discretization issues of rating-based methods, where responses are typically binary or lie on a 1–5 star scale (Davenport et al., 2014); and (iii) comparing two items is cognitively easier than rating them on an abstract scale (Stewart et al., 2005). In fact, the advantages of ordinal (comparison) feedback over cardinal (rating) feedback have been empirically demonstrated in small-scale tasks (Shah et al., 2016). We ask whether it is possible to learn personalized preferences from such comparison data in a computationally and statistically efficient manner.

To answer this question, it is natural to consider the following comparison-based recommender system model that arises by combining the matrix completion assumptions with a discrete choice model. Specifically, assume each user $u$ has a utility $x_{u,i}$ for every item $i$, where $x_{u,i}$ is defined as the inner product of a low-dimensional user feature vector $u$ and item feature vector $v$. Thus the utility matrix $X$ can be factorized into feature matrices as follows: $X = UV^T$. Comparisons follow a noisy oracle: when presented with two items $i$ and $j$, the user $u$ picks $i$ over $j$ with probability

---

[1]School of Computer and Communication Sciences, EPFL, Switzerland [2]Department of Computer Science, TU Munich, Germany. Correspondence to: Suryanarayana Sankagiri <suryanarayana.sankagiri@epfl.ch>.

*Proceedings of the 42$^{nd}$ International Conference on Machine Learning*, Vancouver, Canada. PMLR 267, 2025. Copyright 2025 by the author(s).

$g(x_{u,i} - x_{u,j})$ for some known link function $g(\cdot)$. Given a dataset that is generated from this model with some ground-truth features $(U^*, V^*)$, one can learn these features by optimizing the likelihood of the model over the dataset. These learned features can then be used to predict each user's preferences over unseen items. The key theoretical question of interest is whether there exists an efficient algorithm that can provably recover the ground-truth features.

There has been some progress towards this question, notably by Park et al. (2015) and Negahban et al. (2018). They prove sample complexity guarantees, showing that it is indeed possible to estimate the ground-truth features, even when each user makes only a few comparisons. However, their analysis rests on a convex problem posed in terms of the entire utility matrix $X$, and is therefore inefficient to solve. In contrast, the optimization problem posed in terms of the feature matrices $(U, V)$ is computationally much easier to solve, despite being nonconvex. Indeed, the nonconvex approach has been applied to large-scale, real-world comparison datasets, where the convex approach would be infeasible (Rendle et al., 2009; Park et al., 2015). While this approach is computationally faster, the nonconvexity in the problem makes it harder to provide theoretical guarantees. Proving recovery guarantees for this nonconvex approach has been highlighted as an important open problem by Negahban et al. (2018). Our work aims to plug this gap in the literature.

In this work, we provide the first theoretical recovery guarantees for the nonconvex learning-from-comparisons problem. Our guarantees stem from a careful analysis of the loss landscape. We show that within a neighborhood of the true solution, the negative log-likelihood function exhibits a strong convexity-like property. Therefore, with a warm start, (projected) gradient descent converges exponentially fast to the global minimum (see Theorem 4.1). Our proof of this result involves two broad steps. First, we demonstrate that the *expected* log likelihood function satisfies this desirable structural property. Second, we show that the *empirical* log likelihood function is close to its expected value, even when the dataset is sparse. Showing the latter requires strong concentration results. Our work introduces new techniques to establish these results, extending the methods developed for the matrix completion problem. Our guarantees are qualitatively similar to those established for matrix completion, notably the work of Zheng & Lafferty (2016). Further details of our technical contributions are presented after a review of related literature.

Our simulation results corroborate our theoretical findings. Importantly, they show that some of the assumptions (such as the warm start) needed for our theoretical result are not necessary in practice. They also show that the constants in our theoretical results—the sample complexity and the convergence rate—are quite conservative. Our work suggests that explicitly asking users to compare pairs of items (instead of rating them) can be a viable approach to learning user preferences. To the best of our knowledge, such a dataset is not publicly available. Note that prior works using near-identical methods on real datasets, such as Rendle et al. (2009) and Park et al. (2015), infer comparisons from other forms of data. We hope our work will motivate practitioners to collect a large comparison dataset similar to the Netflix dataset, on which our method can be tested. An interesting direction of future research is to empirically test whether comparison-based feedback leads to better recommendations than rating-based feedback, possibly due to lower noise.

## 2. RELATED WORK

### 2.1. On Matrix Completion

The matrix completion problem can be stated as follows: recover a low-rank matrix given a small subset of its entries, possibly corrupted by noise. There are two approaches that provide theoretically optimal solutions for this problem. One approach involves posing a nuclear norm minimization program, subject to the constraints that some select entries must match the observation. Candès & Recht (2009) was the first to theoretically establish that a low rank matrix can be recovered exactly, given a small, randomly sampled, subset of its entries (without noise). Later work extended these results to the setting with noisy observations (Candes & Plan, 2010; Negahban & Wainwright, 2012).

An alternate approach is to pose the problem in its *matrix factorization* form. This is based on the observation that a low-rank matrix $X$ admits a factorization into two smaller matrices $(U, V)$: $X = UV^T$. One can pose a squared-loss minimization problem in terms of the factors $(U, V)$ (Mnih & Salakhutdinov, 2007). While this alternate formulation leads to a nonconvex objective function, it is much faster to solve and yields good results on real data (Koren et al., 2009). This empirical success spurred a long line of theoretical research on analyzing this nonconvex optimization problem. The work of Keshavan et al. (2010a) was the first to provide theoretical guarantees for this nonconvex formulation. They propose a two-step method. First, they show that performing the singular value decomposition of the partially observed matrix leads to a candidate solution close to the ground-truth. Next, using this matrix as a starting point, they show that a gradient-descent like method converges to the true solution.

Other works have built upon this initial result to show slightly stronger theoretical guarantees with improved proof techniques (Chen & Wainwright, 2015; Sun & Luo, 2016; Zheng & Lafferty, 2016). Following the two-step approach

prescribed by Keshavan et al. (2010a), all these works focus on proving that there exists a basin of attraction around the true solution. Notably, all these papers use a key concentration result developed by Candès & Recht (2009) (Theorem 4.1). This result, in turn, relies on the following assumptions (i) the ground-truth matrix is *incoherent* (no row or column of the matrix dominates the rest) and (ii) the observed entries are chosen uniformly at random from all the entries of the matrix. Furthermore, these methods require that the iterates always remain incoherent throughout; to this end, they use a regularizer (Sun & Luo, 2016) or a projection step (Chen & Wainwright, 2015; Zheng & Lafferty, 2016).

Further work on this problem has led to significant relaxations in the assumptions needed to prove theoretical guarantees. Firstly, Ge et al. (2016) and Ge et al. (2017) show that all local minima are global in the nonconvex formulation. This implies gradient-based methods are guaranteed to converge to a global optimum, even without the initialization procedure. Secondly, Ma et al. (2020) shows that gradient descent has implicit regularization and thus can converge to the optimal solution without an explicit regularizer or a projection operation.

### 2.2. On Learning From Comparisons

The central problem in learning from comparison data is to estimate the preference order/rank of all the items given a dataset. A popular approach to solve this problem is to assume the comparisons arise from a probabilistic choice model, such as the Bradley-Terry-Luce choice model. Theoretical guarantees for learning the parameters of this model have been established in the literature (Negahban et al., 2012; Maystre & Grossglauser, 2015; Shah et al., 2016). In addition to the offline setting, the corresponding active learning problem has also been well-studied, especially in the framework of dueling bandits (Bengs et al., 2021). In particular, the contextual dueling bandit model is quite similar to our model; however, with the 'collaborative filtering' aspect missing, the basic estimation problem there reduces to that of logistic regression (Saha, 2021; Bengs et al., 2022).

The problem of learning a low-rank user-item score matrix from comparison data was first formulated by Rendle et al. (2009). This work applied a model and algorithm very similar to ours to a comparison dataset derived from implicit user feedback such as views, clicks, and purchases. Rendle et al. (2009) demonstrated that such data is better treated as ordinal information (a preference of the viewed item over the rest) instead of cardinal information (a positive rating of the viewed item).

Park et al. (2015) was the first work to provide theoretical guarantees for this problem, albeit for convex version. This work also noted the similarity of this problem to the matrix completion setting, prompting them to also develop a more

efficient nonconvex method which is nearly identical to ours. They applied this method to a comparison dataset derived from movie ratings (higher rated movie is preferred over a lower rated one), getting recommendations of a quality similar to processing ratings directly, thereby establishing the efficacy of this method.

Negahban et al. (2018) studies this problem in much greater detail, providing matrix recovery guarantees with optimal sample complexity. It also analyzes more complex settings such as sampling item pairs in a nonuniform fashion and learning from one-out-of-$k$ choices. Ultimately, the paper focuses only on the convex formulation, stating that the analysis of the corresponding nonconvex formulation is an important open problem.

### 2.3. Our Technical Contributions

In this work, we provide a theoretical analysis of the nonconvex formulation for the problem of learning a low-rank matrix from comparison data. The modeling assumptions we make, such as the incoherence of the ground-truth matrix and the uniform sampling of datapoints, are very similar to prior work on matrix factorization. Our proof strategy is also inspired by prior work on this subject; most notably, that of Zheng & Lafferty (2016). In particular, we follow their approach of using a regularizer to translate an asymmetric matrix factorization problem ($X = UV^T$) into a symmetric one ($Y = ZZ^T$). We also follow their idea of using projected gradient descent to ensure the iterates stay incoherent.

The key difference between our work and prior work is the method used to develop the necessary concentration inequalities. Most of the papers analyzing matrix completion build upon some fundamental results from Candès & Recht (2009) and Keshavan et al. (2010a).However, these results do not apply to our problem, because the structure of the *sampling matrix* is different. To elaborate, in matrix completion, a data point consists of a single user and a single item, while here, a datapoint consists of a single user and an item-pair. This seemingly minor difference makes us lose the interpretation of the set of samples acting like a projection operator (Candès & Recht, 2009), or the samples being edges of a bipartite graph (Keshavan et al., 2010a). In this work, we derive the necessary concentration results by using the matrix Bernstein inequality (Tropp, 2015) as the main tool. Further details are given in Section 5.

We make two major simplifying assumptions in this work. First, we assume that our comparisons are noiseless. That is, instead of observing a binary preference outcome, we observe the expected value of this outcome. Extending our analysis to the more realistic setting of noisy, binary com-

parisons is an important direction of future work.[1] Second, we assume we are given an initial point that is suitably close to the ground truth solution. In the matrix completion literature, such an initial solution can be obtained by performing a singular value decomposition on the partially observed matrix, as shown by (Keshavan et al., 2010a). However, this initialization method does not work here. Our simulations in Section 6 suggest that this warm start may not be a necessity. Proving convergence from a random point, as done by (Ge et al., 2016), remains an open problem.

## 3. Model

### 3.1. The Data Generation Process

Let there be a fixed set of users and items. Let $n_1$ denote the number of users and $n_2$ the number of items. We assume that each user $u$ has a certain utility $x^*_{u,i}$ for every item $i$. Each user $u$ and each item $i$ has a $r$-dimensional feature vector. Further, we assume that the score matrix $X^* \in \mathbb{R}^{n_1 \times n_2}$ has rank $r$. Although our analysis holds for any rank $r \leq \min n_1, n_2$, the results are interesting when the matrix is low rank, *i.e.,* the $r$ is much smaller than $n_1$ and $n_2$. We call $X^*$ the ground-truth utility matrix.

When a user $u$ is asked to choose which option they prefer between two items $i$ and $j$, we assume the user makes a choice by comparing their utilities. To be precise, let $w = 1$ denote the event that the user picks $i$ over $j$ and $w = 0$ the complementary event. We assume

$$P(w = 1) = g(x^*_{u,i} - x^*_{u,j}), \tag{1}$$

where $g : \mathbb{R} \to (0, 1)$ is called the *link function*. This model captures the intuition that the user is certain in their choices among items that differ significantly in their utility, but is more ambiguous when choosing between similar utility items. A special case is the classical Plackett-Luce choice model, where the link function is the sigmoid function: $g(x) = 1/(1 + \exp(-x))$. In general, the link function is a smooth, strictly increasing function and is symmetric around zero in the following sense: $g(-x) = 1 - g(x)$.

We assume we are given a dataset $\mathcal{D}$ where each data point represents a comparison made by a user between two items. The size of the dataset, *i.e.,* the number of data points, is represented by $m$. We index the dataset by $k$. Each data point $\mathcal{D}_k$ is of the form $((u; i, j), w)$ and is sampled randomly as follows. The user index $u$ is chosen uniformly at random from $[n_1]$. The pair of item indices $(i, j)$ is chosen uniformly at random from the set of $n_2(n_2 - 1)$ pairs of

---

[1]Indeed, in the matrix completion literature as well, the noiseless case has been addressed first and the noisy case in a follow up work (e.g., Candès & Recht (2009) followed by Candes & Plan (2010), Keshavan et al. (2010a) followed by Keshavan et al. (2010b)).

distinct items. The item pair $(i, j)$ is sampled independently from $u$. The triplets for different datapoints are sampled independently of each other.

In this work, we assume that comparisons are *noiseless*, *i.e.,* instead of a binary outcome, we observe the expected value of the comparison outcome $w$ (that is, $g(x^*_{u,i} - x^*_{u,j})$). Although this assumption is not a reflection of practice, we make this assumption for the simplicity of exposition. By making this assumption, we can show that with sufficient data, we can estimate the ground-truth matrix to arbitrary precision. The binary outcome setting can be viewed as a *noisy* setting, as any random variable $w$ can be expressed as the sum of its mean $\mathbb{E}[w]$ and some mean-zero noise. We believe it is possible to extend our results to the noisy case, except that the recovery guarantees will contain a residual estimation error due to the noise.

### 3.2. Notation

We now introduce some additional notation that we will use throughout the rest of this paper. This notation is useful not only to succinctly represent the loss function (see Section 3.3), but also to argue about the desired concentration results (see Section 5).

In Section 3.1, we assumed that the score matrix $X^* \in \mathbb{R}^{n_1 \times n_2}$ has rank $r$. This implies that it admits the following rank-$r$ SVD: $X^* = U^* \Sigma^* V^{*T}$. Here, $U^* \in \mathbb{R}^{n_1 \times r}$ and $V^* \in \mathbb{R}^{n_2 \times r}$ are orthonormal matrices (satisfying $U^{*T} U^* = V^{*T} V^* = I_r$), and $\Sigma^* \in \mathbb{R}^{r \times r}$ is a diagonal matrix with entries $\sigma^*_1 \geq \ldots \geq \sigma^*_r > 0$.

Let $n = n_1 + n_2$. Define $Z^* \in \mathbb{R}^{n \times r}$ and $Y^* \in \mathbb{R}^{n \times n}$ as follows:

$$Z^* = \begin{bmatrix} U^* \\ V^* \end{bmatrix} \Sigma^{*1/2}, \tag{2}$$

$$Y^* = Z^* Z^{*T} = \begin{bmatrix} U^* \Sigma^* U^{*T} & X^* \\ X^{*T} & V^* \Sigma^* V^{*T} \end{bmatrix}. \tag{3}$$

We can interpret $U^* \Sigma^{*1/2}$ as the matrix of user feature vectors, with row $u$ corresponding to user $u$. Similarly, $V^* \Sigma^{*1/2}$ can be viewed the matrix of item feature vectors. Both user and item features are $r$-dimensional vectors. Note that $X^* = (U^* \Sigma^{*1/2})(V^* \Sigma^{*1/2})^T$. Thus, the user-item utility $x^*u, i$ can be viewed as the inner product of the corresponding user and item feature vectors.

Given the relation between matrices $X^*$, $Y^*$, and $Z^*$, estimating the ground-truth utility matrix $X^*$ is equivalent to estimating $Z^*$ (barring the symmetries discussed in Section 3.5). The major advantage of this reduction is that it reduces the number of parameters from $n_1 n_2$ (in $X^*$) to $(n_1 + n_2)r$ (in $Z^*$). This significant reduction in parameters (when $r$ is small) leads to corresponding gains in computational efficiency. Thus, from here on, the goal of the learning

problem is to estimate $Z^*$. Before we present the precise learning problem in subsequent subsections, we introduce some more notation that will make the presentation concise.

Let $e_1, e_2, \ldots e_{n_1}$ denote unit vectors in $\mathbb{R}^{n_1}$ and let $\tilde{e}_1, \tilde{e}_2, \ldots, \tilde{e}_{n_2}$ denote unit vectors in $\mathbb{R}^{n_2}$. Let $\langle\!\langle C, D \rangle\!\rangle = \sum_{i,j} c_{i,j} d_{i,j}$ denote the matrix inner product between two matrices of the same size. Therefore:

$$\langle\!\langle e_u(\tilde{e}_i - \tilde{e}_j)^T, X^* \rangle\!\rangle = x^*_{u,i} - x^*_{u,j}. \tag{4}$$

For any triplet $(u; i, j)$, define the corresponding *sampling matrix* $A \in \mathbb{R}^{n \times n}$ to be:

$$A = \begin{bmatrix} 0 & e_u(\tilde{e}_i - \tilde{e}_j)^T \\ 0 & 0 \end{bmatrix}. \tag{5}$$

In the equation above, $0$ denotes matrices with all entries zero of the appropriate size. With this notation, for any data point $((u; i, j), w)$, we have:

$$\langle\!\langle A, Y^* \rangle\!\rangle = \langle\!\langle A^T, Y^* \rangle\!\rangle = x^*_{u,i} - x^*_{u,j} \tag{6}$$
$$\Rightarrow P(w = 1 \,|\, (u; i, j)) = P(w = 1 \,|\, A) = g(\langle\!\langle A, Y^* \rangle\!\rangle).$$

Let $A_k$ denote the sampling matrix corresponding to the datapoint $\mathcal{D}_k$. By our modeling assumptions above, $A_1, A_2, \ldots$ are i.i.d. random matrices of the same form as (5), with the index $u$ being chosen uniformly at random from $[n_1]$, and the indices $(i, j)$ being chosen uniformly at random from $[n_2]$ (with the condition $i \neq j$).

Lastly, for any matrix $Y \in \mathbb{R}^{n \times n}$, define

$$\mathcal{D}(Y) \triangleq \frac{1}{m} \sum_{k=1}^{m} \langle\!\langle A_k + A_k^T, Y \rangle\!\rangle^2. \tag{7}$$

We overload this notation to highlight the fact that the operator $\mathcal{D}(\cdot)$ captures the collective action of all the sampling matrices of the dataset $\mathcal{D}$. We shall encounter such terms repeatedly in our analysis. Observe that $\mathcal{D}(Y)$ is the empirical mean of i.i.d. random terms. Thus, it is reasonable to expect that $\mathcal{D}(Y) \approx \mathbb{E}[\mathcal{D}(Y)]$, if the number of datapoints $m$ is sufficiently large. Our analysis rests on proving such concentration results; see Section 5 for more details.

### 3.3. The Loss Function

Recall, from the previous section, that our goal is to estimate the matrix $Z^*$. We do so by maximizing the likelihood as a function of matrices $Z \in \mathbb{R}^{n \times r}$ over the dataset $\mathcal{D}$. In other words, we formulate a loss function in terms of the negative log likelihood, and minimize this function using a gradient descent-like method. This section presents the expressions for the log likelihood and its gradient, using the notation developed in the previous section.

Given a binary outcome $w$, the likelihood of the outcome under a Bernoulli distribution with parameter $p$ is

$p^w(1-p)^{1-w}$. Therefore, the negative log-likelihood of this observation is $-w \log(p) - (1-w)\log(1-p)$. Next, consider a datapoint $((u; i, j), w)$ with the corresponding sampling matrix $A$. The negative log-likelihood of this observation with parameters $Z$ is

$$-w \log(g(\langle\!\langle A, ZZ^T \rangle\!\rangle)) - (1-w)\log(1 - g(\langle\!\langle A, ZZ^T \rangle\!\rangle)).$$

Then, for the entire dataset, the (normalized) negative log likelihood is given by:

$$\mathcal{L}(Z) = \frac{1}{m} \sum_{k=1}^{m} -w_k \log(g(\langle\!\langle A_k, ZZ^T \rangle\!\rangle))$$
$$- (1 - w_k)\log(1 - g(\langle\!\langle A_k, ZZ^T \rangle\!\rangle)). \tag{8}$$

The gradient of $\mathcal{L}(Z)$ is

$$\nabla \mathcal{L}(Z) = \frac{1}{m} \sum_{k=1}^{m} h_k (A_k + A_k^T) Z, \text{ where} \tag{9}$$

$$h_k \triangleq \frac{g'(z_k)(g(z_k) - w_k)}{g(z_k)(1 - g(z_k))}, \quad z_k \triangleq \langle\!\langle A_k, ZZ^T \rangle\!\rangle.$$

Here, $\nabla \mathcal{L}(Z)$ is a matrix of the same size as $Z$ while $h_k$ and $z_k$ are scalars. Finally, note that with the noiseless assumption, we can substitute $w_k$ by $g(\langle\!\langle A_k, Z^* Z^{*T} \rangle\!\rangle)$.

### 3.4. Important Parameters

**Condition Number** Let $\sigma_1^*, \sigma_2^*, \ldots \sigma_r^*$ denote the singular values of $X^*$. Denote the ratio $\sigma_1^*/\sigma_r^*$, called the *condition number* of the data, by $\kappa$. Also note $\kappa$ is also the condition number of $Z^*$, because the singular values of $Z^*$ are $\sqrt{2\sigma_1^*}, \sqrt{2\sigma_2^*}, \ldots \sqrt{2\sigma_r^*}$.

**Incoherence** For any matrix $Z$, let $\|Z\|_{2,\infty}$ denote the maximum of the $\ell_2$ norm of its rows and let $\|Z\|_F$ denote the Frobenius norm of $Z$. Define the *incoherence parameter* of the ground-truth matrix as

$$\mu \triangleq n(\|Z^*\|_{2,\infty}^2 / \|Z^*\|_F^2). \tag{10}$$

In principle, $\mu$ can take values from $1$ to $n$. However, the sample complexity worsens with $\mu$, as the concentration bounds weaken with $\mu$.

**Link Function Bounds** Let $I$ denote the interval $[-24\mu(\|Z^*\|_F^2/n), 24\mu(\|Z^*\|_F^2/n)]$. Let $\xi$ and $\Xi$ be lower and upper bounds for the following expression:

$$\xi \triangleq \min_{x \in I, y \in I} \frac{g'(x)g'(y)}{g(x)(1 - g(x))}, \tag{11}$$

$$\Xi \triangleq \max_{x \in I, y \in I} \frac{g'(x)g'(y)}{g(x)(1 - g(x))}. \tag{12}$$

By the assumptions on $g(\cdot)$ stated above, $\xi$ is strictly positive and $\Xi$ is finite. These terms are used to bound the term $h_k$ in (9) above.

## 3.5. Symmetries in the Problem

The generative model, and consequently the log likelihood function, is invariant to certain transformations in the parameters. We explore these symmetries and their consequences in this section.

**Scale Invariance**   For any score matrix $X$, the mapping to $Z = (U, V)$ is not unique. Indeed, for any invertible $r \times r$ matrix $P$, the matrix $Z' = (UP^T, VP^{-1})$ is indistinguishable from $(U, V)$, as they both lead to the same score matrix $X$ and hence the same likelihood. However, we can distinguish 'imbalanced' matrices from 'balanced' ones by by adding a regularizer term $\left\| U^T U - V^T V \right\|_F^2$ to the loss function. Minimizing this regularizer while keeping the log-likelihood constant leads to a pair of feature matrices that are balanced in the norms. In more compact terms, the regularizer can be written as follows:

$$\mathcal{R}(Z) \triangleq \left\| Z^T D Z \right\|_F^2 ; \ D \triangleq \begin{bmatrix} I_{n_1} & 0 \\ 0 & -I_{n_2} \end{bmatrix}. \qquad (13)$$

Note that the ground-truth matrix $Z^*$ satisfies $\mathcal{R}(Z^*) = 0$. Combining the regularizer with the negative log likelihood, the objective function becomes:

$$f(Z) \triangleq \mathcal{L}(Z) + (\lambda/4)\mathcal{R}(Z), \qquad (14)$$

where $\lambda$ is a positive constant. In this work, we set $\lambda = \xi\gamma/4$; however, in practice, it should be treated as a hyperparameter. In summary, adding the regularizer $\mathcal{R}(Z)$ factors out the scale-invariance of the problem.

**Rotational Invariance**   Beyond the scale invariance, the problem at hand also exhibits rotational invariance. Let $R$ be any orthogonal matrix in $r$ dimensions, *i.e.*, $R \in \mathbb{R}^{r \times r}$ such that $RR^T = R^T R = I$. The matrix $ZR = (UR, VR)$ give rise to the same scores as $Z = (U, V)$. Thus, one can identify the ground-truth features only up to an orthogonal transformation. Denote this equivalence class of the ground-truth feature matrices by $\Phi$:

$$\Phi \triangleq \{ \tilde{Z}^* : \tilde{Z}^* = Z^* R \text{ for some orthonormal } R \}. \quad (15)$$

This equivalence class of solutions naturally gives rise to a new distance metric $\Delta$ that measures how close a candidate solution $Z$ is to $\Phi$. Define

$$R(Z) \triangleq \arg \min_{R : R^T R = RR^T = I_r} \left\| Z - Z^* R \right\|_F, \qquad (16)$$

$$\Phi(Z) \triangleq \arg \min_{\tilde{Z}^* \in \Phi} \left\| Z - \tilde{Z}^* \right\|_F = Z^* R(Z), \qquad (17)$$

$$\Delta(Z) \triangleq Z - \Phi(Z). \qquad (18)$$

We measure the quality of a solution $Z$ by $\|\Delta(Z)\|_F$.

**Shift Invariance**   Under our model, all comparisons involve computing the difference between the utilities of two items. Therefore, adding a constant vector to each item's feature vector does not affect the scores. Mathematically, this can be seen as follows. Let $\tilde{V}^* = V^* + 1v^T$, where $v \in \mathbb{R}^r$ and $1 \in \mathbb{R}^{n_2}$ is the vector of all ones. Let $\tilde{X}^*, \tilde{Y}^*$, and $\tilde{Z}^*$ denote the corresponding quantities derived from $(U^*, \tilde{V}^*)$. Then for any triplet $(u; i, j)$ and the corresponding sampling matrix $A$, we have $\langle\!\langle e_u(\tilde{e}_i - \tilde{e}_j)^T, \tilde{X}^* \rangle\!\rangle = \langle\!\langle e_u(\tilde{e}_i - \tilde{e}_j)^T, X^* \rangle\!\rangle$, which implies $\langle\!\langle A, \tilde{Y}^* \rangle\!\rangle = \langle\!\langle A, Y^* \rangle\!\rangle$. Because of this invariance, we assume, without loss of generality, that $1^T V^* = 0$. In words, we assume that the item features of all matrices in $\Phi$ sum to zero.

The shift invariance also manifests itself in our objective function $\mathcal{L}(Z)$. It is important to factor out the shift invariance in order to establish a strong-convexity like property (i.e., a curvature) for $\mathcal{L}(Z)$. Therefore, we restrict our attention to the following subspace:

$$\mathcal{H} = \{ Z \in \mathbb{R}^{n \times r} : Z = (U, V), 1^T V = 0 \}. \qquad (19)$$

For any $Z = (U, V)$, we shall work with the projection of $Z$ onto $\mathcal{H}$, denoted by $\mathcal{P}_{\mathcal{H}}(Z)$. This projection is given by $(U, JV)$, where $J \triangleq I_{n_2} - 11^T/(n_2)$. Finally, note that by the assumption stated before, $\Phi \subseteq \mathcal{H}$.

# 4. Algorithm and Result

A naive approach to minimize the loss function (14) is to simply apply the gradient descent method until one is sufficiently close to convergence. Indeed, in Section 6, we show this works well in practice. However, for proving theoretical guarantees, we need to use *projected gradient descent*. Notably, the projection step involves two successive projections, first onto a set of 'incoherent matrices' $\mathcal{C}$ and then onto $\mathcal{H}$ (defined in (19)). The set $\mathcal{C}$ is defined as follows:

$$\mathcal{C} \triangleq \left\{ Z \in \mathbb{R}^{n \times r} : \|Z\|_{2,\infty} \leq \frac{4}{3} \sqrt{\frac{\mu}{n}} \left\| Z^0 \right\|_F \right\}. \quad (20)$$

Thus, $\mathcal{C}$ contains matrices that are 'nearly as incoherent' as $Z^*$ (if $\left\| Z^0 \right\|_F \approx \|Z^*\|_F$). For any $Z \in \mathbb{R}^{n \times r}$, the projection of $Z$ onto $\mathcal{C}$, $\mathcal{P}_{\mathcal{C}}(Z)$, is a matrix in $\mathbb{R}^{n \times r}$ obtained by clipping the rows of $Z$ to $\beta = (4/3)\sqrt{(\mu/n)} \left\| Z^0 \right\|_F$:

$$\forall j \in [n], \ \mathcal{P}_{\mathcal{C}}(Z)_j = \begin{cases} Z_j & \text{if } \|Z_j\|_2 \leq \beta \\ Z_j(\beta/ \|Z_j\|_2) & \text{otherwise} \end{cases}.$$

The rationale for the projections is the following. One, the objective function displays a strong-convexity like property only within the region of incoherent matrices. The projection operation $\mathcal{P}_{\mathcal{C}}$ ensures that we stay in this region, which is crucial for proving the theoretical results. The second projection, $\mathcal{P}_{\mathcal{H}}$ factors out the shift invariance in the loss

function. This is essential in order to establish strong convexity; otherwise, there is no curvature in the direction of invariance. Here, there is a caveat: this second projection may push the iterates out of the set $\mathcal{C}$. However, we show (in Lemma A.3) that the iterates remain incoherent enough, *i.e.*, they remain in the set $\overline{\mathcal{C}}$, where

$$\overline{\mathcal{C}} \triangleq \left\{ Z \in \mathbb{R}^{n \times r} : \|Z\|_{2,\infty} \leq \sqrt{12\mu/n} \, \|Z^*\|_F \right\} \quad (21)$$

---

**Algorithm 1** Projected Gradient Descent

---

**Input:** Objective function $f$, initial solution $Z^0 \in \mathbb{R}^{n \times r}$, stepsize $\eta$
$t \leftarrow 0$
$Z^0 \leftarrow \mathcal{P}_{\mathcal{H}} \left( \mathcal{P}_{\mathcal{C}} \left( Z^0 \right) \right)$
**repeat**
 $Z^{t+1} \leftarrow \mathcal{P}_{\mathcal{H}} \left( \mathcal{P}_{\mathcal{C}} \left( Z^t - \eta \nabla f(Z^t) \right) \right)$
 $t \leftarrow t + 1$
**until** convergence
**Output:** $Z^t$

---

Our main theorem states that in the noiseless setting, given a sufficiently large dataset and a warm start, Algorithm 1 converges exponentially fast to a solution equivalent to the ground-truth matrix. For the sake of conciseness, we introduce the following constants:

$$\gamma \triangleq 2/(n_1(n_2 - 1)), \quad \tau \triangleq \xi/\Xi, \quad \alpha \triangleq \xi\gamma\sigma_r^*.$$

Let $\mathcal{B}(\varepsilon) = \{Z : \|\Delta(Z)\|_F^2 \leq \varepsilon\sigma_r^*\}$ denote a 'ball' around the true solution. With this notation in place, we state the main result of this paper.

**Theorem 4.1.** *Suppose the following conditions hold:*

- *The dataset $\mathcal{D}$ consists of $m$ i.i.d. samples generated according to the model presented in Section 3.1.*

- *The number of samples $m$ is at least $10^7 \left(\mu r \kappa/\tau\right)^2 n \log\left(8n/\delta\right)$ for some $\delta \in (0, 1)$.*

- *The initial point $Z^0$ lies in $\mathcal{B}(\tau/50)$.*

- *The stepsize $\eta$ in Algorithm 1 satisfies $\eta\alpha \leq 2.5 \cdot 10^{-6}(\tau/\mu r \kappa)^2$.*

*Then, with probability at least $1 - \delta$, the iterates $Z^1, Z^2, \ldots$ of Algorithm 1 satisfy:*

$$\left\|\Delta(Z^t)\right\|_F^2 \leq \left(1 - \frac{\alpha\eta}{4}\right)^t \left\|\Delta(Z^0)\right\|_F^2 \quad \forall t \in \mathbb{N}.$$

We highlight two important points from the above theorem. First, the dependence on the problem size is $O(nr^2 \log n)$, which is near-optimal. Second, for a well-chosen stepsize, the algorithm convergences exponentially at rate

$O((\tau/\mu r \kappa)^2)$. Although the constants in the sample complexity result and convergence rate are quite large in the statement of Theorem 4.1, our experimental results in Section 6 show that in practice, these constants are moderate. The following section gives a sketch of the proof of Theorem 4.1. The full proof is provided in the appendix.

## 5. Proof Outline

Theorem 4.1 is nearly identical to the convergence guarantees of gradient descent for a strongly convex and smooth function, notwithstanding the projection step and the symmetries. Lemmas 5.1 and 5.2 establish properties akin to strong-convexity and smoothness respectively. Note that we have dropped the dependence on $Z$ for brevity; *e.g.*, we denote $\nabla f(Z)$ by $\nabla f$.

**Lemma 5.1.** *Suppose the number of samples $m$ is at least $10^7 \left(\mu r \kappa/\tau\right)^2 n \log\left(2n/\delta\right)$, for some $\delta \in (0, 1)$. Then, with probability at least $1 - \delta$, $\forall\, Z \in \mathcal{H} \cap \mathcal{B}(\tau/50) \cap \overline{\mathcal{C}}$,*

$$\langle\!\langle \nabla f, \Delta \rangle\!\rangle \geq \frac{\xi\gamma}{4} \|\Delta\|_F^2 + \frac{\xi\gamma}{8} \left\|\Delta^T D\Phi\right\|_F^2.$$

**Lemma 5.2.** *Suppose the number of samples $m$ is at least $2n \log(4n/\delta)$, for some $\delta \in (0, 1)$. Then, with probability at least $1 - \delta$, $\forall\, Z \in \mathcal{B}(1) \cap \overline{\mathcal{C}}$,*

$$\|\nabla f\|_F^2 \leq 10^5 (\Xi\gamma\mu r\sigma_1^*)^2 \|\Delta\|_F^2 + \frac{(\xi\gamma)^2}{2}\sigma_1^* \left\|\Phi^T D\Delta\right\|_F^2.$$

The lemmas are easy to interpret as strong convexity and smoothness conditions if we ignore the terms $\left\|\Delta^T D\Phi\right\|_F^2$ (which stem from the regularizer).

At a high level, the method for proving both these lemmas is similar. First, the expressions to be bounded, namely $\langle\!\langle \nabla f, \Delta \rangle\!\rangle$ and $\|\nabla f\|_F^2$, are written out as the sum and product of $\mathcal{D}(Y)$ terms (recall the definition of $\mathcal{D}(Y)$ from (7)). Second, we demonstrate that these terms, which capture an empirical mean of i.i.d. random variables, are close to their statistical mean. Specifically, we show that with high probability, $\mathcal{D}(Y) \approx \mathbb{E}[\mathcal{D}(Y)]$, uniformly for all $Y$ in some appropriate set. Finally, we put these results together with the appropriate parameters to ensure that the bounds presented in Lemmas 5.1 and 5.2 hold. The following sections flesh out more details.

**Showing Strong Convexity** The proof of Lemma 5.1 can be split into the following three lemmas.

**Lemma 5.3.** *For any $Z \in \overline{\mathcal{C}}$,*

$$\langle\!\langle \nabla \mathcal{L}, \Delta \rangle\!\rangle \geq \frac{\xi}{2}\mathcal{D}\left(\Delta\Phi^T\right) - \frac{5\Xi}{8}\mathcal{D}\left(\Delta\Delta^T\right)$$

**Lemma 5.4.** *Let some $\epsilon, \delta \in (0,1)$ be given. Suppose the number of samples $m$ exceeds $96\mu r \left(\kappa/\epsilon\right)^2 n \log\left(n/\delta\right)$. Then, with probability at least $1 - \delta$, $\forall Z \in \mathcal{H}$,*

$$\mathcal{D}\left(\Delta\Phi^T\right) \geq \gamma \left((1-\epsilon)\sigma_r^* \|\Delta\|_F^2 + 2\langle\!\langle \Phi_U \Delta_V^T, \Delta_U \Phi_V^T \rangle\!\rangle\right).$$

In the above lemma, we use the notation $\Phi = (\Phi_U, \Phi_V)$ and $\Delta = (\Delta_U, \Delta_V)$.

**Lemma 5.5.** *Let some $\epsilon, \delta \in (0,1)$ be given. Suppose the number of samples $m$ exceeds $845 \left(\mu r \kappa/\epsilon\right)^2 n \log\left(n/\delta\right)$. Then, with probability at least $1 - \delta$, $\forall Z \in \overline{\mathcal{C}} \cap \mathcal{B}(\epsilon)$,*

$$\mathcal{D}(\Delta\Delta^T) \leq 10\epsilon\gamma\sigma_r^* \|\Delta\|_F^2.$$

Using these three lemmas, Lemma 5.1 can be derived in a straightforward manner (proof in Appendix E). Indeed, if we ignore the cross-term $\langle\!\langle \Phi_U \Delta_V^T, \Delta_U \Phi_V^T \rangle\!\rangle$ in Lemma 5.4, it is not hard to see that the three lemmas combined lead to the lower bound $\langle\!\langle \nabla\mathcal{L}, \Delta \rangle\!\rangle \geq O(1)\gamma\sigma_r^* \|\Delta\|_F^2$. The gradient of the regularizer helps cancel out this cross-term, but leads to the additional $\|\Delta D\Phi\|_F^2$ term.

The steps in the proof of Lemma 5.3 are algebraic in nature and largely follow the pattern presented in Zheng & Lafferty (2016); the proof is given in Appendix B. The main technical contribution of our work lies in the proof of Lemmas 5.4 and 5.5. Although the statements of these lemmas are similar to Lemmas 10 and 8 respectively of Zheng & Lafferty (2016), we prove these results in different ways. We outline the broad steps taken to prove these results, filling in the details in Appendices C and D respectively.

A key step to prove Lemma 5.4 is to show the identity:

$$\mathcal{D}\left(\Delta\Phi^T\right) = v^T S_{\mathcal{D}} v, \text{ where } v \triangleq \mathsf{vec}(\Delta R^T),$$
$$S_{\mathcal{D}} \triangleq \frac{1}{m} \sum_{k=1}^m a_k a_k^T, \ a_k \triangleq \mathsf{vec}((A_k + A_k^T)Z^*). \quad (22)$$

Here, we use the notion of vectorization of a matrix, *i.e.*, stacking the columns of a matrix to form a vector. Thus, for a matrix $Z \in \mathbb{R}^{n \times r}$, $\mathsf{vec}(Z)$ is a vector in $\mathbb{R}^{nr}$.

Given this quadratic form, it follows that:

$$\left| \mathcal{D}\left(\Delta\Phi^T\right) - \mathbb{E}\left[\mathcal{D}\left(\Delta\Phi^T\right)\right] \right| \leq \|S_{\mathcal{D}} - \mathbb{E}[S_{\mathcal{D}}]\|_2 \|v\|_2^2$$

The term $\|S_{\mathcal{D}} - \mathbb{E}[S_{\mathcal{D}}]\|_2$ can be bounded with high probability using the matrix Bernstein inequality (see Lemma C.6). To complete the proof of Lemma 5.4, it remains to calculate $\mathbb{E}\left[\mathcal{D}\left(\Delta\Phi^T\right)\right]$. In Lemma C.2, we show that $\mathbb{E}\left[\mathcal{D}\left(\Delta\Phi^T\right)\right] = \gamma \|\Delta_U \Phi_V^T + \Phi_U \Delta_V^T\|_F^2$.

The proof of Lemma 5.5, just like the one for Lemma 5.4, involves analyzing a quadratic form around a random matrix,

which we split into the mean (expectation) term and the deviation from the mean. We show that:

$$\mathcal{D}(\Delta\Delta^T) = y^T B_{\mathcal{D}} y = y^T \mathbb{E}[B_{\mathcal{D}}]y + y^T (B_{\mathcal{D}} - \mathbb{E}[B_{\mathcal{D}}])y;$$
$$y \in \mathbb{R}^n : y_j = \|\Delta_j\|_2^2 \ \forall j, \ B_{\mathcal{D}} = \frac{1}{m} \sum_{(u;i,j)\in\mathcal{D}} e_u(\tilde{e}_i + \tilde{e}_j).$$

The first term is bounded above with the warm-start assumption: $\|\Delta\|_F^2 \leq O(1)\sigma_r^*$. The second term is bounded using the matrix Bernstein inequality (see Lemma D.5).

**Showing Smoothness** Our method of proving Lemma 5.2 follows the proof style of Zheng & Lafferty (2016). We start by observing that

$$\|\nabla\mathcal{L}\|_F^2 = \sup_{W \in \mathbb{R}^{n \times r} : \|W\|_F = 1} \langle\!\langle \nabla\mathcal{L}, W \rangle\!\rangle^2.$$

Therefore, it suffices to find a bound for the term on the right hand side of the above equation. The following lemmas, proven in Appendix D, provide the requisite bound.

**Lemma 5.6.** *For any $Z \in \overline{\mathcal{C}}$ and any $W \in \mathbb{R}^{n \times r}$,*

$$\langle\!\langle \nabla\mathcal{L}, W \rangle\!\rangle^2 \leq 2\Xi^2 \left(\mathcal{D}(\Delta\Phi^T) + \frac{1}{4}\mathcal{D}(\Delta\Delta^T)\right) \mathcal{D}(WZ^T).$$

**Lemma 5.7.** *Suppose the number of samples $m$ is at least $2n \log(4n/\delta)$. Then, with probability at least $1 - \delta$, the following inequalities hold uniformly for all $Z \in \overline{\mathcal{C}}$:*

$$\mathcal{D}(\Delta\Phi^T) \leq 16\gamma(\mu r \sigma_1^*) \|\Delta\|_F^2,$$
$$\mathcal{D}(\Delta\Delta^T) \leq 416\gamma(\mu r \sigma_1^*) \|\Delta\|_F^2,$$
$$\mathcal{D}(WZ^T) \leq 192\gamma(\mu r \sigma_1^*) \|W\|_F^2 \ \forall W \in \mathbb{R}^{n \times r}.$$

Lemma 5.2 follows by combining these lemmas and accounting for the gradient of the regularizer (see Appendix E).

# 6. Simulations

**Data Generation:** We generated a random ground truth matrix $X^* \in \mathbb{R}^{n_1 \times n_2}$ with entries selected independently at random according to normal distribution and calculated its rank-$r$ SVD, $U^* \Sigma^* V^{*T}$. We have two settings: a low-dimensional setting with $(n_1, n_2) = (200, 300)$ and a high-dimensional setting with $(n_1, n_2) = (2000, 3000)$. In both settings, we had $r = 3, \mu \approx 1.01, \kappa = 1.1$. Using this matrix, we randomly and independently collected $m$ comparison data points. Specifically, for each setting, the comparison dataset took the form $\{(A_k, w_k) : k = 1, \ldots, m\}$, where $A_k$ represents the $k^{\text{th}}$ sampling matrix as in (5) and and $w_k = g(\langle\!\langle A_k, Z^* Z^{*T} \rangle\!\rangle)$. In this work, we set the regularizer coefficient to be $\lambda = \gamma/40$. Subsequently, we applied Algorithm 1 using the stepsize $\eta$ as recommended

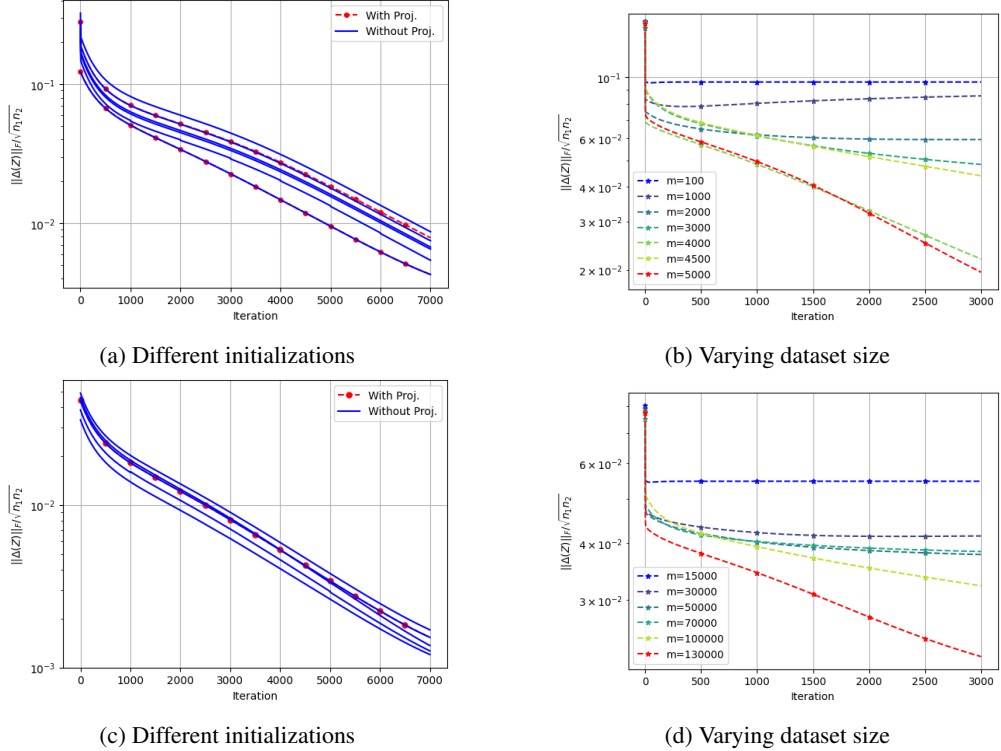

(a) Different initializations

(b) Varying dataset size

(c) Different initializations

(d) Varying dataset size

*Figure 1.* The top row and bottom row show the results for $(n_1, n_2) = (200, 300)$ and $(n_1, n_2) = (2000, 3000)$, respectively. (a) and (c) illustrate the effect of different initializations with a fixed number of data points, while the remaining plots demonstrate the effect of varying dataset size $m$. Y-axes are in log scale.

by Theorem 4.1. The quality of the algorithm's output at iteration $t$ is measured by $\|\Delta(Z^t)\|_F / \sqrt{n_1 n_2}$. Our code can be found here. Figure 1 presents the resulting plots.

**Initialization:** We initialize the algorithm with $Z^0 = Z^{*T} + \vartheta(N_1, N_2 J)$, where $N_1 \in \mathbb{R}^{n_1 \times r}$ and $N_2 \in \mathbb{R}^{n_2 \times r}$, with their entries drawn from a standard normal distribution. For our experiments, we use $\vartheta \in \{0.5, 1, 2\}$. Figures 1 (a) and (c) show the effect of different initial solutions and also the projection steps in low and high dimensional settings, respectively. In both settings, the number of data points $m$ and also the stepsize were chosen as recommended in Theorem 4.1. This result confirms the linear convergence of Algorithm 1 as predicted by our theoretical analysis. It is important to emphasize that while both a warm start and the projection step are required for our theoretical guarantees, these simulation results suggest that they are not needed in practice.

**Dataset size:** We examine the impact of dataset size $m$ on the algorithm's performance. Figures 1 (b) and (d) demonstrate the resulting normalized errors in low and high dimensional settings, respectively. As depicted in these plots, a large enough $m$ leads to linear convergence of the algorithm while for a small $m$, the error $\|\Delta(Z^t)\|_F$ does not go to zero as $t$ increases. In both plots, the red curves show the con-

verges rate for $m$ computed by $c_0(\mu r \kappa)^2 n \log(n/\delta)$ with $\delta = 0.05$ and $c_0$ being $1/4$ for low-dimensional and $1/2$ for high-dimensional setting. In Appendix F, we present an additional plot that highlights the dependence of $m$ on $r$.

## 7. Conclusion

In this paper, we consider a mathematical model for a comparison-based recommender system: the concatenation of the classical matrix factorization framework with a Plackett-Luce-style comparison oracle. We proved that, given a relatively sparse dataset, the parameters of the model can be recovered through an efficient, gradient descent based algorithm, despite the loss function being nonconvex. Our proof rests on establishing that the loss function satisfies properties akin to strong convexity and smoothness in a neighborhood around the optimal solution. For our analysis, we made two assumptions: we are given a warm start and we observe the exact choice probabilities (rather than binary outcomes). We hope that our work will form the basis of further analysis of this problem that performs a global analysis or provides guarantees for data with noisy comparisons. Finally, we believe that this work is an important contribution in establishing the viability of comparison-based recommender systems.

## Acknowledgements

This research has been supported in part by the Swiss National Science Foundation (SNSF) under grant `IZBRZ2_186313`.

## Impact Statement

This paper presents work whose goal is to advance the field of Machine Learning. There are many potential societal consequences of our work, none which we feel must be specifically highlighted here.

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

# A. Helper Lemmas

## A.1. Matrix Inner Product Identities

We state some basic identities of the matrix inner product operator, which are trivial to verify but are used frequently in the paper. In the following identities, $D, E,$ and $F$ are arbitrary matrices so long as their sizes are compatible with the equations.

$$\langle\langle E, F \rangle\rangle = \mathsf{Tr}(EF^T) = \mathsf{Tr}(FE^T) \tag{23}$$

$$\langle\langle E, F \rangle\rangle = \langle\langle F, E \rangle\rangle = \langle\langle E^T, F^T \rangle\rangle \tag{24}$$

$$\langle\langle DE, F \rangle\rangle = \langle\langle D, FE^T \rangle\rangle = \langle\langle E, D^T F \rangle\rangle, \quad \langle\langle D, EF \rangle\rangle = \langle\langle DF^T, E \rangle\rangle = \langle\langle E^T D, F \rangle\rangle \tag{25}$$

From these identities, we get that for any sampling matrix $A$ (defined in (5)) and any $Y, Z \in \mathbb{R}^{n \times r}$:

$$\begin{aligned} \langle\langle (A + A^T)Y, Z \rangle\rangle &= \langle\langle AY, Z \rangle\rangle + \langle\langle A^T Y, Z \rangle\rangle \\ &= \langle\langle A, ZY^T \rangle\rangle + \langle\langle A^T, ZY^T \rangle\rangle \\ &= \langle\langle A, ZY^T + YZ^T \rangle\rangle \end{aligned} \tag{26}$$

Let $W$ and $Z$ be two matrices in $\mathbb{R}^{n \times r}$. Recall the notation convention introduced in Section 5. Using the above identity and (5), we get that for any sampling matrix $A$ corresponding to the triplet $(u; i, j)$,

$$\langle\langle (A + A^T), WZ^T \rangle\rangle = \langle\langle e_u(\tilde{e}_i - \tilde{e}_j)^T, W_U Z_V + Z_U W_V \rangle\rangle \tag{27}$$

$$= \langle\langle W_u, Z_i - Z_j \rangle\rangle + \langle\langle Z_u, W_i - W_j \rangle\rangle \tag{28}$$

## A.2. The Frobenius Norm of the Product of Two Matrices

Let $X$ be any matrix and let $\sigma_{\max}(X)$ and $\sigma_{\min}(X)$ denote the largest and smallest singular values of $X$. Let $v$ be any vector such that the product $Xv$ is compatible. By the definition of singular values:

$$\sigma_{\min}(X) \|v\|_2 \leq \|Vx\|_2 \leq \sigma_{\max}(X) \|v\|_2$$

Using this basic fact, we can prove the following result.

**Lemma A.1.** *Let $U \in \mathbb{R}^{n_1 \times r}$ and $V \in \mathbb{R}^{n_2 \times r}$ be any two matrices. Let $\sigma_1(U) \geq \ldots \geq \sigma_r(U)$ denote the singular values of $U$ and $\sigma_1(V) \geq \ldots \geq \sigma_r(V)$ denote the singular values of $V$. Then $\|UV^T\|_F^2$ satisfies the following bounds:*

$$\sigma_r(U)^2 \|V\|_F^2 \leq \|UV^T\|_F^2 \leq \sigma_1(U)^2 \|V\|_F^2$$

$$\sigma_r(V)^2 \|U\|_F^2 \leq \|UV^T\|_F^2 \leq \sigma_1(V)^2 \|U\|_F^2$$

*Proof.* We first prove the inequality $\|UV^T\|_F^2 \geq \sigma_r(U)^2 \|V\|_F^2$. Let $V_j$ denote the $j^{\text{th}}$ row of $V$, written as a column vector ($r \times 1$ matrix). Let $(UV^T)^j$ denote the $j^{\text{th}}$ column of $UV^T$. Finally, note that the squared Frobenius norm of a matrix is the sum of the squared $\ell_2$ norms of its rows or of its columns. Stitching together these simple facts, we get.

$$\begin{aligned} \|UV^T\|_F^2 &= \sum_{j=1}^{n_2} \|(UV^T)^j\|_2^2 = \sum_{j=1}^{n_2} \|UV_j\|_2^2 \\ &\geq \sum_{j=1}^{n_2} \sigma_r(U)^2 \|V_j\|_2^2 = \sigma_r(U)^2 \sum_{j=1}^{n_2} \|V_j\|_2^2 \\ &= \sigma_r(D)^2 \|V\|_F^2 \end{aligned}$$

The upper bound $\|UV^T\|_F^2 \leq \sigma_1(U)^2 \|V\|_F^2$ can be derived using the same steps, except we use the inequality $\|UV_j\|_2 \leq \sigma_1(U) \|V_j\|_2$ instead of $\|UV_j\|_2 \geq \sigma_r(U) \|V_j\|_2$. Finally, the second set of bounds follow by applying the first set of bounds to the matrix $VU^T$, and noting that $\|UV^T\|_F = \|VU^T\|_F$. $\square$

### A.3. The Incoherence of the Iterates

Recall that we have assumed that the initial point $Z^0$ satisfies the bound $\left\|\Delta(Z^0)\right\|_F^2 \leq \sigma_r^*/16$, i.e., we are given a warm start (see Section 4). With this assumption, we can prove the following lemmas.

**Lemma A.2.** *Let $\mathcal{C}$ be the set defined in (20), i.e.,*

$$\mathcal{C} \triangleq \left\{ Z \in \mathbb{R}^{n \times r} \ : \ \|Z\|_{2,\infty} \leq \frac{4}{3}\sqrt{\frac{\mu}{n}} \left\|Z^0\right\|_F \right\}$$

*Then all the equivalent ground-truth matrices lie in $\mathcal{C}$, i.e. $\Phi \subseteq \mathcal{C}$.*

*Proof.* Start with the identity $Z^0 = \Phi(Z^0) + \Delta(Z^0)$ (which follows from (18)). By the triangle inequality, we get

$$\left\|\Phi(Z^0)\right\|_F - \left\|\Delta(Z^0)\right\|_F \leq \left\|Z^0\right\|_F \leq \left\|\Phi(Z^0)\right\|_F + \left\|\Delta(Z^0)\right\|_F .$$

Note that all matrices in $\Phi$ have the same Frobenius norm. This implies that $\left\|\Phi(Z^0)\right\|_F = \|Z^*\|_F$. Combining this with the bound on $\left\|\Delta(Z^0)\right\|_F$, we get

$$\|Z^*\|_F - \sqrt{\sigma_r^*}/4 \leq \left\|Z^0\right\|_F \leq \|Z^*\|_F + \sqrt{\sigma_r^*}/4 \tag{29}$$

Recall that the singular values of $Z^*$ are $\sqrt{2\sigma_1^*}, \sqrt{2\sigma_2^*}, \dots, \sqrt{2\sigma_r^*}$. We know that the Frobenius norm of a matrix is the $\ell_2$ norm of the vector of its singular values. Therefore:

$$\|Z^*\|_F = \sqrt{2 \sum_{i=1}^{r} \sigma_i^*} \Rightarrow \frac{\sqrt{\sigma_r^*}}{4} \leq \frac{\|Z^*\|_F}{4}$$

$$\Rightarrow \left\|Z^0\right\|_F \geq \|Z^*\|_F - \sqrt{\sigma_r^*}/4 \geq \frac{3}{4}\|Z^*\|_F$$

$$\Rightarrow \|Z^*\|_{2,\infty} = \sqrt{\mu/n}\|Z^*\|_F \leq \frac{4}{3}\sqrt{\mu/n}\left\|Z^0\right\|_F$$

Thus, we see that $Z^* \in \mathcal{C}$. Because all $Z \in \Phi$ have the same $\ell_2/\ell_\infty$ norm, it follows that $\Phi \subseteq \mathcal{C}$. $\qquad\square$

Before proceeding further, we introduce some new notation. Recall the convention (established in Section 5) that any matrix $Z$ can be viewed as a concatenation of two matrices: $Z = (Z_U, Z_V)$. To index the rows of $Z$, we use $Z_u, u \in [n_1]$ for the user features and $Z_i, Z_j, j \in [n_2]$ for the item features. In expressions involving matrix multiplication, we view $Z_u, Z_i, Z_j$ as row vectors, i.e., as $1 \times r$ matrices. By the definition of $\|Z\|_{2,\infty}$, we get:

$$\|Z\|_{2,\infty} = \max\{ \max_{u \in [n_1]} \|Z_u\|_2 \,, \max_{i \in [n_2]} \|Z_i\|_2 \}. \tag{30}$$

Equipped with this new notation, we can state and prove the next result.

**Lemma A.3.** *For any $Z \in \mathcal{C}$, let $W = \mathcal{P}_{\mathcal{H}}(Z)$. Then $W \in \overline{\mathcal{C}}$, i.e., $W$ satisfies*

$$\|W\|_{2,\infty}^2 \leq \frac{12\mu}{n}\|Z^*\|_F^2$$

*Proof.* Let $z$ denote the mean of the rows of $Z_V$, i.e.,

$$z \triangleq \frac{1}{n_2} \sum_{i \in [n_2]} Z_i$$

It follows that

$$\Rightarrow \|z\|_2 = \frac{1}{n_2} \left\| \sum_{i \in [n_2]} Z_i \right\|_2 \leq \frac{1}{n_2} \sum_{i \in [n_2]} \|Z_i\|_2 \leq \frac{1}{n_2} \sum_{i \in [n_2]} \|Z\|_{2,\infty} = \|Z\|_{2,\infty} \quad \text{(by (30))}$$

The operation of projecting onto the subspace $\mathcal{H}$ is such that $W_U = Z_U$ and $W_i = Z_i - v$ for all item rows $i$ (see Section 3.5). By the triangle inequality, we get:

$$\|W_i\|_2 = \|Z_i - z\|_2 \leq \|Z_i\|_2 + \|z\|_2$$
$$\Rightarrow \max_{i \in [n_2]} \|W_i\|_2 \leq \max_{i \in [n_2]} \|Z_i\|_2 + \|z\|_2 \leq \|Z\|_{2,\infty} + \|z\|_2 \leq 2 \|Z\|_{2,\infty}$$

Because the rows of $U$ remain unchanged, we have $\|W\|_{2,\infty} \leq 2 \|Z\|_{2,\infty}$.

Next, note that $Z \in \mathcal{C}$. Therefore,

$$\|Z\|_{2,\infty} \leq \frac{4}{3}\sqrt{\frac{\mu}{n}} \left\|Z^0\right\|_F \leq \frac{5}{3}\sqrt{\frac{\mu}{n}} \|Z^*\|_F$$

The last step uses the inequality $\left\|Z^0\right\|_F \leq (5/4) \|Z^*\|_F$, which follows from (29) in the derivation of Lemma A.2. By combining the above inequalities, we get the desired result:

$$\left\|\hat{Z}\right\|_{2,\infty}^2 \leq 4 \|Z\|_{2,\infty}^2 \leq 4\frac{25}{9}\frac{\mu}{n} \|Z^*\|_F^2 \leq \frac{12\mu}{n} \|Z^*\|_F^2 .$$

$\square$

The above result is important because it establishes a useful bound that holds for all iterates $Z^t, t \in \mathbb{Z}_+$. (Recall that Algorithm 1 takes successive projections, first on to $\mathcal{C}$ and then onto $\mathcal{H}$.)

### A.4. Bounds on the Scores

In this subsection, we derive two related bounds on any $Z \in \mathbb{R}^{n \times r}$ and any sampling matrix $A$:

$$|\langle\!\langle A, ZZ^T \rangle\!\rangle| \leq 2 \|Z\|_{2,\infty}^2 \tag{31}$$

$$\left\|(A + A^T)Z\right\|_F^2 \leq 6 \|Z\|_{2,\infty}^2 \tag{32}$$

Before we prove these bounds, let us explore its consequence. By the definition of the incoherence parameter $\mu$ (10), $\|Z^*\|_{2,\infty}^2 = (\mu/n) \|Z^*\|_F^2$. Therefore,

$$|\langle\!\langle A, Z^*Z^{*T} \rangle\!\rangle| \leq \frac{2\mu}{n} \|Z^*\|_F^2 \tag{33}$$

$$\left\|(A + A^T)Z^*\right\|_F^2 \leq \frac{6\mu}{n} \|Z^*\|_F^2 \tag{34}$$

Moreover, for all $Z \in \overline{\mathcal{C}}$,

$$|\langle\!\langle A, ZZ^T \rangle\!\rangle| \leq \frac{24\mu}{n} \|Z^*\|_F^2 \tag{35}$$

$$\left\|(A + A^T)Z\right\|_F^2 \leq \frac{72\mu}{n} \|Z^*\|_F^2 \tag{36}$$

As argued in the previous subsection, all iterates $(Z^t)_{t \in \mathbb{Z}_+}$ of Algorithm 1 lie in $\overline{\mathcal{C}}$ and consequently satisfy the above bound.

We now proceed to the derivation of (31). Let $Z \in \mathbb{R}^{n \times r}$ be some candidate feature matrix and let $X = Z_U Z_V^T$ be the corresponding score matrix. Let $(u; i, j)$ be an arbitrary triplet and let $A$ denote the corresponding sampling matrix. Recall the definition of the sampling matrix $A$ corresponding to a triplet $(u; i, j)$ from (4) and (5). We have

$$|\langle\!\langle A, ZZ^T \rangle\!\rangle| = |x_{u,i} - x_{u,j}| = |\langle Z_u, (Z_i - Z_j)\rangle| \leq \|Z_u\|_2 \|Z_i - Z_j\|_2 \leq \|Z_u\|_2 (\|Z_i\|_2 + \|Z_j\|_2) \leq 2 \|Z\|_{2,\infty}^2$$

The last inequality follows from the definition of $\|Z\|_{2,\infty}$ (see (30)).

The derivation of (32) proceeds as follows.

$$A = \begin{bmatrix} 0 & e_u(\tilde{e}_i - \tilde{e}_j)^T \\ 0 & 0 \end{bmatrix}$$

$$\Rightarrow A + A^T = \begin{bmatrix} 0 & e_u(\tilde{e}_i - \tilde{e}_j)^T \\ (\tilde{e}_i - \tilde{e}_j)e_u^T & 0 \end{bmatrix}$$

$$\Rightarrow (A + A^T)Z = \begin{bmatrix} 0 & e_u(\tilde{e}_i - \tilde{e}_j)^T \\ (\tilde{e}_i - \tilde{e}_j)e_u^T & 0 \end{bmatrix} \begin{bmatrix} Z_U \\ Z_v V \end{bmatrix}$$

$$= \begin{bmatrix} e_u(\tilde{e}_i - \tilde{e}_j)^T Z_V \\ (\tilde{e}_i - \tilde{e}_j)e_u^T Z_U \end{bmatrix}$$

$$= \begin{bmatrix} e_u(Z_i - Z_j) \\ (\tilde{e}_i - \tilde{e}_j)Z_u \end{bmatrix}$$

$$\Rightarrow \left\| (A + A^T)Z \right\|_F^2 = \left\| e_u(Z_i - Z_j) \right\|_F^2 + \left\| (\tilde{e}_i - \tilde{e}_j)Z_u \right\|_F^2$$

$$= \left\| e_u \right\|_2^2 \left\| Z_i - Z_j \right\|_2^2 + \left\| \tilde{e}_i - \tilde{e}_j \right\|_2^2 \left\| Z_u \right\|_2^2$$

$$= \left\| Z_i - Z_j \right\|_2^2 + 2 \left\| Z_u \right\|_2^2 \quad (\left\| e_u \right\|_2^2 = 1,\ \left\| \tilde{e}_i - \tilde{e}_j \right\|_2^2 = 2)$$

$$\leq 2(\left\| Z_i \right\|_2^2 + \left\| Z_j \right\|_2^2) + 2 \left\| Z_u \right\|_2^2 \quad (\left\| Z_i - Z_j \right\|_2^2 \leq (\left\| Z_i \right\|_2 + \left\| Z_j \right\|_2)^2 \leq 2(\left\| Z_i \right\|_2^2 + \left\| Z_j \right\|_2^2))$$

$$\leq 6 \left\| Z \right\|_{2,\infty}^2 \quad \text{(by definition of } \left\| Z \right\|_{2,\infty} \text{ (30))}$$

This establishes the second inequality.

## A.5. The Matrix Bernstein Inequality

Here, we state a special version of the matrix Bernstein inequality that we use in our proofs. The statement is identical to Corollary 6.2.1 in (Tropp, 2015), barring a change in notation.

This concentration result is stated in terms of the operator norm of a matrix $X$, which we denote as $\left\| X \right\|_2$ and is defined as follows:

$$\left\| X \right\|_2 \triangleq \sup_{v: \left\| v \right\|_2 = 1} \left\| Xv \right\|_2 \tag{37}$$

It follows that $\left\| X \right\|_2 = \sigma_{\max}(X)$. For square matrices $X$, an alternate definition of the operator norm is:

$$\left\| X \right\|_2 \triangleq \sup_{v: \left\| v \right\|_2 = 1} v^T X v \tag{38}$$

**Lemma A.4** (Matrix Bernstein Inequality). *Consider a random matrix $X$ of shape $n_1 \times n_2$ that satisfies:*

$$\mathbb{E}[X] = \bar{X} \quad \text{and} \quad \left\| X \right\|_2 \leq L \text{ almost surely.}$$

*Let $b$ be an upper bound on the second moment of $X$:*

$$\left\| \mathbb{E}[XX^T] \right\|_2 \leq b \quad \text{and} \quad \left\| \mathbb{E}[X^T X] \right\|_2 \leq b.$$

*Let $X_{\mathcal{D}} = \frac{1}{m} \sum_{k=1}^m X_k$, where each $X_k$ is an i.i.d. copy of $X$. Then, for all $t \geq 0$,*

$$P(\left\| X_{\mathcal{D}} - \bar{X} \right\|_2 \geq t) \leq (n_1 + n_2) \exp\left( \frac{-mt^2/2}{b + 2Lt/3} \right)$$

# B. Initial Lemmas

Following the convention of the main paper, we drop the explicit dependence on $Z$ wherever it is obvious.

## B.1. Proof of Lemma 5.3

**Lemma 5.3.** *For any $Z \in \overline{\mathcal{C}}$,*

$$\langle\!\langle \nabla \mathcal{L}, \Delta \rangle\!\rangle \geq \frac{\xi}{2} \mathcal{D}\left(\Delta \Phi^T\right) - \frac{5\Xi}{8} \mathcal{D}\left(\Delta \Delta^T\right)$$

*Proof.* From the expression of $\nabla \mathcal{L}$ (see (9)), we get that:

$$\langle\!\langle \nabla \mathcal{L}, \Delta \rangle\!\rangle = \frac{1}{m} \sum_{k=1}^{m} h_k \langle\!\langle (A_k + A_k^T)Z, \Delta \rangle\!\rangle \text{ where } h_k = \frac{g'(z_k)\left(g(z_k) - w_k\right)}{g(z_k)(1 - g(z_k))}, \ z_k = \langle\!\langle A_k, ZZ^T \rangle\!\rangle.$$

Recall, by definition (see (18)), that $Z = \Phi + \Delta$. Therefore. the term $\langle\!\langle (A_k + A_k^T)Z, \Delta \rangle\!\rangle$ can be expanded as follows:

$$\langle\!\langle (A_k + A_k^T)Z, \Delta \rangle\!\rangle = \langle\!\langle (A_k + A_k^T)\Phi, \Delta \rangle\!\rangle + \langle\!\langle (A_k + A_k^T)\Delta, \Delta \rangle\!\rangle$$
$$= \langle\!\langle A_k + A_k^T, \Delta \Phi^T \rangle\!\rangle + \langle\!\langle A_k + A_k^T, \Delta \Delta^T \rangle\!\rangle \quad \text{(by (25))}$$

Since we have assumed that our observations are noiseless, we have the identity $w_k = g(\langle\!\langle A_k, Z^* Z^{*T} \rangle\!\rangle)$. Plugging this equation in the expression of $h_k$, we get:

$$h_k = \frac{g'(z_k)\left(g(z_k) - g(z_k^*)\right)}{g(z_k)(1 - g(z_k))}; \quad z_k^* = \langle\!\langle A_k, Z^* Z^{*T} \rangle\!\rangle = \langle\!\langle A_k, \Phi \Phi^T \rangle\!\rangle$$

By the mean value theorem,

$$g(z_k) - g(z_k^*) = g'(y_k)(z_k - z_k^*) \quad \text{for some } y_k \text{ in the interval between } z_k \text{ and } z_k^*$$
$$= g'(y_k)\left(\langle\!\langle A_k, ZZ^T \rangle\!\rangle - \langle\!\langle A_k, \Phi \Phi^T \rangle\!\rangle\right)$$
$$= g'(y_k)\left(\langle\!\langle A_k, \Phi \Delta^T + \Delta \Phi^T \rangle\!\rangle + \langle\!\langle A_k, \Delta \Delta^T \rangle\!\rangle\right) \quad \text{(because } Z = \Phi + \Delta)$$
$$= g'(y_k)\left(\langle\!\langle A_k + A_k^T, \Delta \Phi^T \rangle\!\rangle + \frac{1}{2}\langle\!\langle A_k + A_k^T, \Delta \Delta^T \rangle\!\rangle\right) \quad \text{(by (26))}$$

Putting the above equations together, we get:

$$h_k \langle\!\langle (A_k + A_k^T)Z, \Delta \rangle\!\rangle$$
$$= \frac{g'(z_k)g'(y_k)}{g(z_k)(1 - g(z_k))}\left(\langle\!\langle A_k + A_k^T, \Delta \Phi^T \rangle\!\rangle + \frac{1}{2}\langle\!\langle A_k + A_k^T, \Delta \Delta^T \rangle\!\rangle\right)\left(\langle\!\langle A_k + A_k^T, \Delta \Phi^T \rangle\!\rangle + \langle\!\langle A_k + A_k^T, \Delta \Delta^T \rangle\!\rangle\right)$$
$$= \frac{g'(z_k)g'(y_k)}{g(z_k)(1 - g(z_k))}\left(\langle\!\langle A_k + A_k^T, \Delta \Phi^T \rangle\!\rangle^2 + \frac{3}{2}\langle\!\langle A_k + A_k^T, \Delta \Phi^T \rangle\!\rangle\langle\!\langle A_k + A_k^T, \Delta \Delta^T \rangle\!\rangle + \frac{1}{2}\langle\!\langle A_k + A_k^T, \Delta \Delta^T \rangle\!\rangle^2\right)$$
$$\geq \frac{g'(z_k)g'(y_k)}{g(z_k)(1 - g(z_k))}\left(\frac{1}{2}\langle\!\langle A_k + A_k^T, \Delta \Phi^T \rangle\!\rangle^2 - \frac{5}{8}\langle\!\langle A_k + A_k^T, \Delta \Delta^T \rangle\!\rangle^2\right)$$

The last step uses the inequality $2a^2 + 3ab + b^2 \geq a^2 - \frac{5b^2}{4}$, which can be derived from the trivial inequality $(a + 3b/2)^2 \geq 0$. Note also that the coefficient $\frac{g'(z_k)g'(y_k)}{g(z_k)(1 - g(z_k))}$ is positive.

Finally, observe that we have assumed $Z \in \overline{\mathcal{C}}$. The bounds in (33) and (35) imply

$$|z_k^*| \leq 2\frac{\mu \|Z^*\|_F^2}{n}, \ |z_k| \leq 24\frac{\mu \|Z^*\|_F^2}{n}, \text{ which implies } |y_k| \leq 24\frac{\mu \|Z^*\|_F^2}{n}$$

Thus, $y_k$ and $z_k$ lie in the interval $\left[-24\mu \|Z^*\|_F^2 / n, 24\mu \|Z^*\|_F^2 / n\right]$. By the definition of $\xi$ and $\Xi$ in (11) and (12), as well as the definition of the operator $\mathcal{D}(\cdot)$ in (7), the desired expression follows. $\square$

## B.2. Proof of Lemma 5.6

**Lemma 5.6.** *For any $Z \in \overline{\mathcal{C}}$ and any $W \in \mathbb{R}^{n \times r}$,*

$$\langle\!\langle \nabla \mathcal{L}, W \rangle\!\rangle^2 \le 2\Xi^2 \left( \mathcal{D}(\Delta\Phi^T) + \frac{1}{4}\mathcal{D}(\Delta\Delta^T) \right) \mathcal{D}(WZ^T).$$

*Proof.* The proof of this lemma is similar to the proof of Lemma 5.3. One major difference is that we work with terms of the form $\langle\!\langle A + A^T, Y \rangle\!\rangle$ instead of terms $\langle\!\langle A, Y \rangle\!\rangle$.

Following the steps of the proof of Lemma 5.3, we get:

$$\langle\!\langle \nabla \mathcal{L}, H \rangle\!\rangle = \frac{1}{m} \sum_{k=1}^m h_k \langle\!\langle (A_k + A_k^T)Z, H \rangle\!\rangle \text{ where } h_k = \frac{g'(z_k)\,(g(z_k) - w_k)}{g(z_k)(1 - g(z_k))}, \ z_k = \langle\!\langle A_k, ZZ^T \rangle\!\rangle.$$

$$g(z_k) - g(z_k^*) = g'(y_k) \left( \langle\!\langle A_k, \Phi\Delta^T + \Delta\Phi^T \rangle\!\rangle + \langle\!\langle A_k, \Delta\Delta^T \rangle\!\rangle \right) \quad \text{for some } y_k \text{ in the interval between } z_k \text{ and } z_k^*$$

By (24) and (25), we get:

$$\langle\!\langle (A_k + A_k^T)Z, H \rangle\!\rangle = \langle\!\langle A_k + A_k^T, HZ^T \rangle\!\rangle$$

$$\langle\!\langle A_k, \Phi\Delta^T + \Delta\Phi^T \rangle\!\rangle + \langle\!\langle A_k, \Delta\Delta^T \rangle\!\rangle = \langle\!\langle A_k + A_k^T, \Phi\Delta^T \rangle\!\rangle + \frac{1}{2}\langle\!\langle A_k + A_k^T, \Delta\Delta^T \rangle\!\rangle$$

Putting together the equations above, we get:

$$\langle\!\langle \nabla \mathcal{L}, H \rangle\!\rangle = \frac{1}{m} \sum_{k=1}^m \frac{g'(z_k)g'(y_k)}{g(z_k)(1 - g(z_k))} \left( \langle\!\langle A_k + A_k^T, \Phi\Delta^T \rangle\!\rangle + \frac{1}{2}\langle\!\langle A_k + A_k^T, \Delta\Delta^T \rangle\!\rangle \right) \left( \langle\!\langle A_k + A_k^T, HZ^T \rangle\!\rangle \right) \quad (39)$$

Next, we invoke two straightforward inequalities which apply to any sequence of scalars $(a_k)_{k \in [m]}, (b_k)_{k \in [m]}$, and $(c_k)_{k \in [m]}$ with $a_k \ge 0 \ \forall \ k$:

$$\left( \frac{1}{m} \sum_{k=1}^m a_k b_k c_k \right)^2 \le \left( \frac{1}{m} \sum_{k=1}^m a_k b_k^2 \right) \left( \frac{1}{m} \sum_{k=1}^m a_k c_k^2 \right)$$

$$\left( \frac{1}{m} \sum_{k=1}^m a_k b_k^2 \right) \le \left( \max_{k \in [m]} a_k \right) \left( \frac{1}{m} \sum_{k=1}^m b_k^2 \right)$$

The first inequality can be viewed as a form of the Cauchy-Schwarz inequality and the second, a form of Hölder's inequality.

Squaring both sides of the equation in (39) and applying these inequalities with

$$a_k = \frac{g'(z_k)g'(y_k)}{g(z_k)(1 - g(z_k))}, \ b_k = \langle\!\langle A_k + A_k^T, \Phi\Delta^T \rangle\!\rangle + \frac{1}{2}\langle\!\langle A_k + A_k^T, \Delta\Delta^T \rangle\!\rangle, \ c_k = \langle\!\langle A_k + A_k^T, HZ^T \rangle\!\rangle,$$

and observing that $\max_{k \in [m]} a_k \le \Xi$ (using arguments similar to those in Lemma 5.3), we get

$$\langle\!\langle \nabla \mathcal{L}, H \rangle\!\rangle^2 \le \Xi^2 \left( \frac{1}{m} \sum_{k=1}^m (\langle\!\langle A_k + A_k^T, \Phi\Delta^T \rangle\!\rangle + \frac{1}{2}\langle\!\langle A_k + A_k^T, \Delta\Delta^T \rangle\!\rangle)^2 \right) \left( \frac{1}{m} \sum_{k=1}^m \langle\!\langle A_k + A_k^T, HZ^T \rangle\!\rangle^2 \right)$$

$$\le 2\Xi^2 \left( \left( \frac{1}{m} \sum_{k=1}^m \langle\!\langle A_k + A_k^T, \Phi\Delta^T \rangle\!\rangle^2 \right) + \frac{1}{4} \left( \frac{1}{m} \sum_{k=1}^m \langle\!\langle A_k + A_k^T, \Delta\Delta^T \rangle\!\rangle^2 \right) \right) \left( \frac{1}{m} \sum_{k=1}^m \langle\!\langle A_k + A_k^T, HZ^T \rangle\!\rangle^2 \right)$$

$$= 2\Xi^2 \left( \tilde{\mathcal{D}}(\Delta\Phi^T) + \frac{1}{4}\tilde{\mathcal{D}}(\Delta\Delta^T) \right) \tilde{\mathcal{D}}(HZ^T),$$

giving us the bound we want. $\qquad\square$

## C. A Lower Bound for Strong Convexity

In this section, we present the proof of Lemma 5.4, following the approach presented in Section 5. Recall that the goal is to find a lower bound for $\mathcal{D}\left(\Delta\Phi^T\right)$ that holds with high probability. Our approach will be to first derive an expression for $\mathbb{E}\left[\mathcal{D}\left(\Delta\Phi^T\right)\right]$ and then show that $\mathcal{D}\left(\Delta\Phi^T\right)$ is close enough to its expected value. Crucially, we want this result to hold with high probability uniformly for all $Z \in \overline{\mathcal{C}}$.

### C.1. Computing Expectations

Recall the definition of the sampling matrix $A$ corresponding to a triplet $(u; i, j)$ from (4) and (5). In this section, we view the triplet $(u; i, j)$ as a random variable where $u$ is chosen uniformly at random from $[n_1]$ and the pair of item indices $(i, j)$ is chosen uniformly at random from the set of $n_2(n_2 - 1)$ pairs of distinct items, independent from $u$. Consequently, $e_u$ is a random vector in $\mathbb{R}^{n_1}$, $\tilde{e}_i - \tilde{e}_j$ is a random vector in $\mathbb{R}^{n_2}$, and the sampling matrix $A$ is a random matrix in $\mathbb{R}^{n \times n}$. With this interpretation, we can compute:

$$\mathbb{E}[e_u e_u^T] = \frac{1}{n_1} I_{n_1}, \quad \mathbb{E}[(\tilde{e}_i - \tilde{e}_j)(\tilde{e}_i - \tilde{e}_j)^T] = \frac{2}{n_2 - 1} J, \text{ where } J = I_{n_2} - \frac{1}{n_2} 11^T \tag{40}$$

Also recall that $\gamma$ denotes the constant $2/(n_1(n_2 - 1))$.

Using these identities, we can show the following result.

**Lemma C.1.** *For any matrix $X \in \mathbb{R}^{n_1 \times n_2}$,*

$$\mathbb{E}\left[\langle\!\langle e_u(\tilde{e}_i - \tilde{e}_j)^T, X\rangle\!\rangle^2\right] = \gamma \|XJ\|_F^2$$

*Proof.* This proof makes repeated use of the following properties of the trace operator:

- the trace is invariant under cyclic shifts, i.e., $\mathsf{Tr}(ABC) = \mathsf{Tr}(CAB) = \mathsf{Tr}(BCA)$

- the trace of a scalar is the scalar itself

- the trace is a linear operator which commutes with the expectation

We also use the fact that the indices $u$ and $(i, j)$ are independent, so the expectation $\mathbb{E}[\cdot]$ can be decomposed into $\mathbb{E}_{i,j}[\mathbb{E}_u[\cdot]]$.

$$\langle\!\langle e_u(\tilde{e}_i - \tilde{e}_j)^T, X\rangle\!\rangle = \mathsf{Tr}(e_u(\tilde{e}_i - \tilde{e}_j)^T X^T) = \mathsf{Tr}((\tilde{e}_i - \tilde{e}_j)^T X^T e_u) = (\tilde{e}_i - \tilde{e}_j)^T X^T e_u = e_u^T X(\tilde{e}_i - \tilde{e}_j)$$

$$\Rightarrow \langle\!\langle e_u(\tilde{e}_i - \tilde{e}_j)^T, X\rangle\!\rangle^2 = (\tilde{e}_i - \tilde{e}_j)^T X^T e_u e_u^T X(\tilde{e}_i - \tilde{e}_j)$$

$$\Rightarrow \mathbb{E}[\langle\!\langle e_u(\tilde{e}_i - \tilde{e}_j)^T, X\rangle\!\rangle^2] = \mathbb{E}[(\tilde{e}_i - \tilde{e}_j)^T X^T e_u e_u^T X(\tilde{e}_i - \tilde{e}_j)] = \mathbb{E}_{i,j}[\mathbb{E}_u[(\tilde{e}_i - \tilde{e}_j)^T X^T e_u e_u^T X(\tilde{e}_i - \tilde{e}_j)]]$$

$$= \mathbb{E}_{i,j}[(\tilde{e}_i - \tilde{e}_j)^T X^T \mathbb{E}_u[e_u e_u^T] X(\tilde{e}_i - \tilde{e}_j)] = \frac{1}{n_1} \mathbb{E}_{i,j}[(\tilde{e}_i - \tilde{e}_j)^T X^T X(\tilde{e}_i - \tilde{e}_j)] \text{ (by (40))}$$

$$= \frac{1}{n_1} \mathbb{E}_{i,j}[\mathsf{Tr}((\tilde{e}_i - \tilde{e}_j)^T X^T X(\tilde{e}_i - \tilde{e}_j))] = \frac{1}{n_1} \mathbb{E}_{i,j}[\mathsf{Tr}(X^T X(\tilde{e}_i - \tilde{e}_j)(\tilde{e}_i - \tilde{e}_j)^T)]$$

$$= \frac{1}{n_1} \mathsf{Tr}(X^T X \mathbb{E}_{i,j}[(\tilde{e}_i - \tilde{e}_j)(\tilde{e}_i - \tilde{e}_j)^T]) = \frac{2}{n_1(n_2 - 1)} \mathsf{Tr}(X^T X J) \text{ (by (40))}$$

$$= \frac{2}{n_1(n_2 - 1)} \mathsf{Tr}(X^T X J J^T) = \frac{2}{n_1(n_2 - 1)} \mathsf{Tr}((JX)^T X J)$$

$$= \frac{2}{n_1(n_2 - 1)} \|XJ\|_F^2 = \gamma \|XJ\|_F^2$$

In the last but one step, we make use of the fact that $J$ is a projection matrix, which implies $J = JJ^T$. $\qquad\square$

We use Lemma C.1 to prove the next result.

**Lemma C.2.** *For any $Z \in \mathcal{H}$,*

$$\mathbb{E}\left[\mathcal{D}\left(\Delta\Phi^T\right)\right] = \gamma \left\|\Delta_U\Phi_V^T + \Phi_U\Delta_V^T\right\|_F^2,$$

*where $\Phi = (\Phi_U, \Phi_V)$ and $\Delta = (\Delta_U, \Delta_V)$ denote the split of $\Phi$ and $\Delta$ into the first $n_1$ and last $n_2$ rows.*

*Proof.*

$$
\begin{aligned}
\mathbb{E}\left[\mathcal{D}\left(\Delta\Phi^T\right)\right] &= \mathbb{E}\left[\frac{1}{m}\sum_{k=1}^m \langle\!\langle A_k + A_k^T, \Delta\Phi^T \rangle\!\rangle^2\right] \\
&= \mathbb{E}\left[\langle\!\langle A + A^T, \Delta\Phi^T \rangle\!\rangle^2\right] \\
&= \mathbb{E}\left[\langle\!\langle e_u(\tilde{e}_i - \tilde{e}_j)^T, \Phi_U\Delta_V^T + \Delta_U\Phi_V^T \rangle\!\rangle^2\right] \quad \text{(by (27))} \\
&= \gamma \left\|(\Phi_U\Delta_V^T + \Delta_U\Phi_V^T)J\right\|_F^2 \quad \text{(by Lemma C.1)} \\
&= \gamma \left\|\Phi_U\Delta_V^T + \Delta_U\Phi_V^T\right\|_F^2
\end{aligned}
$$

The last step uses the fact that $\Phi_V^T J = \Phi_V^T$ and $\Delta_V^T J = \Delta_V^T$. These identities can be shown as follows. By our assumption on $Z^*$, we know that $Z^* \in \mathcal{H}$. It follows that the entire equivalence class of solutions $\Phi$ lies in $\mathcal{H}$. In particular, $\Phi(Z) \in \mathcal{H}$. We are given some $Z \in \mathcal{H}$. This implies $\Delta(Z) \in \mathcal{H}$, because $\Delta(Z) = Z - \Phi(Z)$ and $\mathcal{H}$ is a vector space. A characterization of $\mathcal{H}$ is that for any $Z = (U, V)$ in $\mathcal{H}$, $JV = V$, or equivalently, $V^T J = V^T$ ($J$ is symmetric). Thus, it follows that $\Phi_V^T J = \Phi_V^T$ and $\Delta_V^T J = \Delta_V^T$. $\qquad\square$

We end this section by bounding the expression in Lemma C.2 from below.

**Lemma C.3.** *For any $Z \in \mathcal{H}$,*

$$\mathbb{E}\left[\mathcal{D}\left(\Delta\Phi^T\right)\right] \geq \gamma \left(\sigma_r^* \|\Delta\|_F^2 + 2\langle\!\langle \Phi_U\Delta_V^T, \Delta_U\Phi_V^T \rangle\!\rangle\right)$$

*Proof.* By Lemma C.2,

$$
\begin{aligned}
\mathbb{E}\left[\mathcal{D}\left(\Delta\Phi^T\right)\right] &= \gamma \left\|\Delta_U\Phi_V^T + \Phi_U\Delta_V^T\right\|_F^2 \\
&= \gamma \left(\left\|\Delta_U\Phi_V^T\right\|_F^2 + \left\|\Phi_U\Delta_V^T\right\|_F^2 + 2\langle\!\langle \Phi_U\Delta_V^T, \Delta_U\Phi_V^T \rangle\!\rangle\right) \\
&\geq \gamma \left(\sigma_r^* \|\Delta_U\|_F^2 + \sigma_r^* \|\Delta_V\|_F^2 + 2\langle\!\langle \Phi_U\Delta_V^T, \Delta_U\Phi_V^T \rangle\!\rangle\right) \quad \text{(by Lemma A.1)} \\
&= \gamma \left(\sigma_r^* \|\Delta\|_F^2 + 2\langle\!\langle \Phi_U\Delta_V^T, \Delta_U\Phi_V^T \rangle\!\rangle\right)
\end{aligned}
$$

Here, we use the fact that $\sigma_r(\Phi_U) = \sigma_r(\Phi_V) = \sqrt{\sigma_r^*}$. This can be shown as follows. Recall $Z^* = (U^*\Sigma^{*1/2}, V^*\Sigma^{*1/2})$ and $\Phi = Z^* R$ for some orthogonal matrix $R$. Therefore, $\Phi_U = U^*\Sigma^{*1/2}R$ and $\Phi_V = V^*\Sigma^{*1/2}R$. These expressions are already in SVD form. Therefore, the singular values for both $\Phi_U$ and $\Phi_V$ are the diagonal elements of $\Sigma^{*1/2}$, namely, $\sqrt{\sigma_1^*}, \ldots, \sqrt{\sigma_r^*}$. $\qquad\square$

## C.2. Vectorization and a Quadratic Form

In this section, we shall show that $\mathcal{D}\left(\Delta\Phi^T\right)$ can be expressed as a quadratic form around a random matrix. This identity will help us prove the desired concentration result in the next section. Let us establish the following notation.

$$v \triangleq \text{vec}(\Delta R^T), \ a_k \triangleq \text{vec}((A_k + A_k^T)Z^*), \ S_k \triangleq a_k a_k^T, \ S_\mathcal{D} \triangleq \frac{1}{m}\sum_{k=1}^m S_k \tag{41}$$

where for any matrix $Z \in \mathbb{R}^{n \times r}$, $\text{vec}(Z)$ is a vector in $\mathbb{R}^{nr}$, obtained by stacking the columns of the matrix one after another. This operation is called 'vectorization of a matrix'. With this notation in place, we proceed to establish the following identities:

**Lemma C.4.**

$$\mathcal{D}\left(\Delta\Phi^T\right) = v^T S_\mathcal{D} v$$
$$\|v\|_2^2 = \|\Delta\|_F^2$$

*Proof.* Recall from (17) that $\Phi = Z^* R$. Let $\tilde{\Delta} \triangleq \Delta R^T$. Then

$$\Delta\Phi^T = \Delta R^T Z^{*T} = \tilde{\Delta} Z^{*T} \Rightarrow \left(\Delta\Phi^T + \Phi\Delta^T\right) = \left(\tilde{\Delta} Z^{*T} + Z^* \tilde{\Delta}^T\right) \tag{42}$$

Next, invoking the notion of vectorization, we get that for any $k \in m$:

$$\langle\!\langle A_k, \tilde{\Delta} Z^{*T} + Z^* \tilde{\Delta}^T \rangle\!\rangle = \langle\!\langle (A_k + A_k^T) Z^*, \tilde{\Delta} \rangle\!\rangle \quad \text{(by (26))}$$
$$= \langle \mathsf{vec}((A_k + A_k^T) Z^*), \mathsf{vec}(\tilde{\Delta}) \rangle$$
$$= \langle a_k, v \rangle \quad \text{(by (7))}$$
$$\therefore \langle\!\langle A_k, \tilde{\Delta} Z^{*T} + Z^* \tilde{\Delta}^T \rangle\!\rangle^2 = \langle v, a_k \rangle \langle a_k, v \rangle$$
$$= v^T S_k v$$
$$\therefore \mathcal{D}\left(\Delta\Phi^T + \Phi\Delta^T\right) = \mathcal{D}\left(\tilde{\Delta} Z^{*T} + Z^* \tilde{\Delta}^T\right) \quad \text{(by (42))}$$
$$= \frac{1}{m} \sum_{i=1}^m \langle\!\langle A_k, \tilde{\Delta} Z^{*T} + Z^* \tilde{\Delta}^T \rangle\!\rangle^2 \quad \text{(by (7))}$$
$$= \frac{1}{m} \sum_{i=1}^m v^T S_k v$$
$$= v^T \left( \frac{1}{m} \sum_{i=1}^m S_k \right) v$$
$$= v^T S_\mathcal{D} v$$

The second statement can be derived easily as shown below:

$$\|v\|_2^2 = \left\|\mathsf{vec}(\Delta R^T)\right\|_2^2$$
$$= \langle \mathsf{vec}(\Delta R^T), \mathsf{vec}(\Delta R^T) \rangle$$
$$= \langle\!\langle \Delta R^T, \Delta R^T \rangle\!\rangle$$
$$= \langle\!\langle \Delta R^T R, \Delta \rangle\!\rangle \quad \text{(by (25))}$$
$$= \langle\!\langle \Delta, \Delta \rangle\!\rangle \quad \text{(because } R \text{ is an orthonormal matrix, } R^T R = I)$$
$$= \|\Delta\|_F^2$$

□

## C.3. A Concentration Result on $S_\mathcal{D}$

Recall, from (41), that $S_\mathcal{D}$ is the empirical mean of i.i.d. random matrices $(S_k)_{k\in m}$. Let $S$ denote the prototype random matrix of which $(S_k)_{k\in[m]}$ are i.i.d. copies, and let $\bar{S}$ denote $\mathbb{E}[S]$. In this section, we will use the matrix Bernstein inequality (Lemma A.4) to establish an upper bound on $\left\|S_\mathcal{D} - \bar{S}\right\|_2$. (Recall from (37) that $\|X\|_2$ denotes the operator norm of $X$.)

In order to apply the matrix Bernstein inequality, we need to compute two parameters, $b$ and $L$, that satisfy:

$$\|S\|_2 \le L \text{ almost surely}, \quad \left\|\mathbb{E}[SS^T]\right\|_2 \le b$$

Here, $S$ is symmetric, so $\mathbb{E}[SS^T] = \mathbb{E}[S^T S]$).

For any rank-one symmetric matrix $Y = yy^T$, $\|Y\|_2 = \|y\|_2^2$. Here, $S = aa^T$ where $a = \mathsf{vec}((A + A^T)Z^*)$ for some sampling matrix $A$. Using this formula, we get

$$\|S\|_2 = \|a\|_2^2 = \left\|\mathsf{vec}((A + A^T)Z^*)\right\|_2^2 = \left\|(A + A^T)Z^*\right\|_F^2 \le \frac{6\mu}{n}\|Z^*\|_F^2 \text{ almost surely (by (34))}$$

Thus, $L = 6(\mu/n)\|Z^*\|_F^2$. Moving on to the calculation for $b$, we get:

$$\mathbb{E}[SS^T] = \mathbb{E}[aa^Taa^T] = \mathbb{E}[\|a\|_2^2\, aa^T] \Rightarrow \left\|\mathbb{E}[SS^T]\right\|_2 = \left\|\mathbb{E}[\|a\|_2^2\, aa^T]\right\|_2 \le \sup_a(\|a\|_2^2)\left\|\mathbb{E}[aa^T]\right\|_2 \le L\left\|\mathbb{E}[aa^T]\right\|_2$$

Where, in the last step, we use the fact that $\|a\|_2^2 \le L$ almost surely. The following lemma establishes the bound $\left\|\mathbb{E}[aa^T]\right\|_2 \le \frac{4\sigma_1^*}{n_1(n_2-1)}$. Thus, we can choose $b = 2\gamma\sigma_1^*L$.

**Lemma C.5.** *Let $a \in \mathbb{R}^{nr}$ denote a random vector such that $a = \mathsf{vec}((A + A^T)Z^*)$, with $A$ being the random sampling matrix defined in Section C.1. Then*

$$\left\|\mathbb{E}[aa^T]\right\|_2 \le 2\gamma\sigma_1^*$$

*Proof.* We adapt the definition of the operator norm of a matrix as follows:

$$\begin{aligned}
\left\|\mathbb{E}[aa^T]\right\|_2 &= \sup_{v \in \mathbb{R}^{nr}:\|v\|_2=1} v^T\mathbb{E}[aa^T]v \\
&= \sup_{Z \in \mathbb{R}^{n \times r}:\|Z\|_F=1} \mathsf{vec}(Z)^T\mathbb{E}[aa^T]\mathsf{vec}(Z) \\
&= \sup_{Z \in \mathbb{R}^{n \times r}:\|Z\|_F=1} \mathbb{E}[\mathsf{vec}(Z)^Taa^T\mathsf{vec}(Z)] \\
&= \sup_{Z \in \mathbb{R}^{n \times r}:\|Z\|_F=1} \mathbb{E}[\langle \mathsf{vec}((A + A^T)Z^*), \mathsf{vec}(Z)\rangle^2] \\
&= \sup_{Z \in \mathbb{R}^{n \times r}:\|Z\|_F=1} \mathbb{E}[\langle\!\langle (A + A^T)Z^*, Z\rangle\!\rangle^2].
\end{aligned}$$

Following the same reasoning as given in the proof of Lemma C.2, we see that:

$$\begin{aligned}
\mathbb{E}[\langle\!\langle (A + A^T)Z^*, Z\rangle\!\rangle^2] &= \gamma\left\|Z_U^*Z_V^TJ + Z_UZ_V^{*T}J\right\|_F^2 \\
&\le \gamma(\left\|Z_U^*Z_V^TJ\right\|_F + \left\|Z_UZ_V^{*T}J\right\|_F)^2 \text{ (by triangle inequality)} \\
&\le 2\gamma(\left\|Z_U^*Z_V^TJ\right\|_F^2 + \left\|Z_UZ_V^{*T}J\right\|_F^2) \text{ (by $(a+b)^2 \le 2(a^2+b^2)$)} \\
&= 2\gamma(\left\|Z_U^*Z_V^TJ\right\|_F^2 + \left\|Z_UZ_V^{*T}\right\|_F^2) \text{ (because $Z_V^{*T}J = Z_V^{*T}$)} \\
&\le 2\gamma(\sigma_1^*\left\|Z_V^TJ\right\|_F^2 + \sigma_1^*\|Z_U\|_F^2) \text{ (by Lemma A.1; $\sigma_1(Z_U^*) = \sigma_1(Z_V^*) = \sqrt{\sigma_1^*}$)} \\
&\le 2\gamma\sigma_1^*(\left\|Z_V^T\right\|_F^2 + \|Z_U\|_F^2) \text{ (by Lemma A.1; $\sigma_1(J) = 1$)} \\
&= 2\gamma\sigma_1^*\|Z\|_F^2
\end{aligned}$$

Plugging this bound into the expression above, we get

$$\begin{aligned}
\left\|\mathbb{E}[aa^T]\right\|_2 &= \sup_{Z \in \mathbb{R}^{n \times r}:\|Z\|_F=1} \mathbb{E}[\langle\!\langle (A + A^T)Z^*, Z\rangle^2] \\
&\le \sup_{Z \in \mathbb{R}^{n \times r}:\|Z\|_F=1} 2\gamma\sigma_1^*\|Z\|_F^2 \\
&= 2\gamma\sigma_1^*
\end{aligned}$$

$\square$

We now have all the ingredients to prove the bound on $\left\|S_{\mathcal{D}} - \bar{S}\right\|_2$.

**Lemma C.6.** *Let $\epsilon \in (0,1)$ and $\delta \in (0,1)$ be given. Suppose the number of samples $m$ is at least $96\mu rn \left(\kappa/\epsilon\right)^2 \log\left(n/\delta\right)$. Then, with probability at least $1 - \delta$,*

$$\left\|S_{\mathcal{D}} - \bar{S}\right\|_2 \leq \gamma \epsilon \sigma_r^*$$

*Proof.* Let the amount of deviation we wish to tolerate be denoted by $t$, *i.e.*, $t = \gamma \epsilon \sigma_r^*$. We have already established the bounds

$$\|S\|_2 \leq L \text{ almost surely}, \quad \left\|\mathbb{E}[SS^T]\right\|_2 \leq b; \quad L = \frac{6\mu}{n}\|Z^*\|_F^2, \ b = 2\gamma \sigma_1^* L$$

Note that $b = (2Lt\kappa/\epsilon)$, since $\kappa = \sigma_1^*/\sigma_r^*$.

By Lemma A.4,

$$P(\left\|S_{\mathcal{D}} - \bar{S}\right\|_2 \geq t) \leq 2nr \exp\left(\frac{-mt^2/2}{b + 2Lt/3}\right)$$

We would like the right hand side to be less than $\delta$. I.e.,

$$2nr \exp\left(\frac{-mt^2/2}{b + 2Lt/3}\right) \leq \delta$$

$$\Leftrightarrow \frac{mt^2/2}{b + 2Lt/3} \geq \log\left(\frac{2nr}{\delta}\right)$$

$$\Leftrightarrow \frac{mt^2/2}{2Lt(\kappa/\epsilon + 1/3)} \geq \log\left(\frac{2nr}{\delta}\right) \quad (\because b = 2Lt\kappa/\epsilon)$$

$$\Leftrightarrow m \geq \frac{4L}{t}\left(\frac{\kappa}{\epsilon} + \frac{1}{3}\right)\log\left(\frac{2nr}{\delta}\right)$$

Next, note that $n = n_1 + n_2$, which implies $n_1(n_2 - 1) \leq n^2$. Further, the Frobenius norm of a matrix is the $\ell_2$ norm of its singular values. We have noted before that the singular values of $Z^*$ are $\sqrt{2\sigma_1^*}, \ldots \sqrt{2\sigma_r^*}$. Therefore $\|Z^*\|_F^2 \leq 2r\sigma_1^*$. Using these inequalities, we get

$$\frac{4L}{t} = 4\left(\frac{6\mu}{n}\|Z^*\|_F^2\right)\left(\frac{1}{\gamma\epsilon\sigma_r^*}\right) = 4\left(\frac{6\mu}{n}\|Z^*\|_F^2\right)\left(\frac{n_1(n_2 - 1)}{2\epsilon\sigma_r^*}\right) = 12\frac{\mu}{\epsilon}\left(\frac{n_1(n_2 - 1)}{n}\right)\left(\frac{\|Z^*\|_F^2}{\sigma_r^*}\right) \leq 24\left(\frac{\mu r\kappa n}{\epsilon}\right)$$

Also note that $\kappa > 1$ and $\epsilon < 1$, so $\kappa/\epsilon + 1/3$ is bounded above by $2\kappa/\epsilon$. Finally, note that $r \leq n_1$ and $r \leq n_2$, so $2r \leq n_1 + n_2 = n$. Therefore, $2nr/\delta \leq n^2/\delta \leq (n/\delta)^2$. Putting these inequalities together, we get:

$$96\mu rn \left(\frac{\kappa}{\epsilon}\right)^2 \log\left(\frac{n}{\delta}\right) \geq \frac{4L}{t}\left(\frac{\kappa}{\epsilon} + \frac{1}{3}\right)\log\left(\frac{2nr}{\delta}\right)$$

Thus, the desired concentration result holds with probability at least $1 - \delta$ if the number of samples $m$ exceeds $96\mu rn \left(\kappa/\epsilon\right)^2 \log\left(n/\delta\right)$. $\qquad \square$

### C.4. Completing the Proof of Lemma 5.4

**Lemma 5.4.** *Let some $\epsilon, \delta \in (0,1)$ be given. Suppose the number of samples $m$ exceeds $96\mu r \left(\kappa/\epsilon\right)^2 n \log\left(n/\delta\right)$. Then, with probability at least $1 - \delta$, $\forall \ Z \in \mathcal{H}$,*

$$\mathcal{D}\left(\Delta\Phi^T\right) \geq \gamma\left((1 - \epsilon)\sigma_r^*\|\Delta\|_F^2 + 2\langle\langle \Phi_U\Delta_V^T, \Delta_U\Phi_V^T \rangle\rangle\right).$$

*Proof.* In Lemma C.4, we established that $\mathcal{D}\left(\Delta\Phi^T\right) = v^T S_{\mathcal{D}} v$. Consequently, $\mathbb{E}[\mathcal{D}\left(\Delta\Phi^T\right)] = v^T \bar{S} v$. Therefore,

$$
\begin{aligned}
|\mathcal{D}\left(\Delta\Phi^T\right) - \mathbb{E}\left[\mathcal{D}\left(\Delta\Phi^T\right)\right]| &= |v^T S_{\mathcal{D}} v - v^T \bar{S} v| \quad \text{(by Lemma C.4)} \\
&= |v^T (S_{\mathcal{D}} - \bar{S}) v| \\
&\leq \left\| S_{\mathcal{D}} - \bar{S} \right\|_2 \left\| v \right\|_2^2 \quad \text{(by (38))} \\
&= \left\| S_{\mathcal{D}} - \bar{S} \right\|_2 \left\| \Delta \right\|_F^2 \quad \text{(by Lemma C.4)} \\
\Rightarrow \mathcal{D}\left(\Delta\Phi^T\right) &\geq \mathbb{E}\left[\mathcal{D}\left(\Delta\Phi^T\right)\right] - \left\| S_{\mathcal{D}} - \bar{S} \right\|_2 \left\| \Delta \right\|_F^2 \\
&\geq \gamma\left(\sigma_r^* \left\| \Delta \right\|_F^2 + 2\langle\!\langle \Phi_U \Delta_V^T, \Delta_U \Phi_V^T \rangle\!\rangle\right) - \left\| S_{\mathcal{D}} - \bar{S} \right\|_2 \left\| \Delta \right\|_F^2 \quad \text{(by Lemma C.3)} \\
&\geq \gamma\left((1-\epsilon)\sigma_r^* \left\| \Delta \right\|_F^2 + 2\langle\!\langle \Phi_U \Delta_V^T, \Delta_U \Phi_V^T \rangle\!\rangle\right) \quad \text{(by Lemma C.6)}
\end{aligned}
$$

$\square$

# D. Upper Bounds for Strong Convexity and Smoothness

## D.1. The Dual Sampling Matrix

Associated with each triplet $(u; i, j)$, we define the *dual sampling matrix* as follows:

$$
B \in \mathbb{R}^{n_1 \times n_2} : B = e_u (\tilde{e}_i + \tilde{e}_j)^T \tag{43}
$$

If we endow the triplets with randomness, $B$ is a random matrix, whose mean is:

$$
\bar{B} \triangleq \mathbb{E}[B] = \mathbb{E}[e_u (\tilde{e}_i + \tilde{e}_j)^T] = \mathbb{E}[e_u]\mathbb{E}[(\tilde{e}_i + \tilde{e}_j)^T] = \frac{2}{n_1 n_2} 11^T \tag{44}
$$

Here, $11^T$ is a matrix of all ones of shape $n_1 \times n_2$.

Let $B_1, \ldots, B_{\mathcal{D}}$ denote the dual sampling matrices for each of the datapoints, similar to the notation for $A$. Define the empirical mean of the dual sampling matrices, $B_{\mathcal{D}}$, as follows:

$$
B_{\mathcal{D}} = \frac{1}{m} \sum_{k=1}^m B_k \tag{45}
$$

In our analysis, we will use the fact that this empirical mean $B_{\mathcal{D}}$ is close to the statistical mean $\bar{B}$, in a manner made precise by Lemma D.5. In preparation for this concentration result, we two parameters, $L$ and $b$. (The same notation was used to denote related terms for the random matrix $S_{\mathcal{D}}$ in the previous section; however, the correct interpretation should be clear from context.) $L$ is a uniform bound on $\|B\|_2$. For each triplet $(u; i, j)$, the operator norm of the corresponding dual sampling matrix is $\sqrt{2}$. It follows that $L = \sqrt{2}$. The definition and bound for $v$ is given in the lemma below.

**Lemma D.1.** *Let $B$ be the random dual sampling matrix as defined above. Let $b^1 \triangleq \left\| \mathbb{E}[B^T B] \right\|_2$, $b^2 \triangleq \left\| \mathbb{E}[BB^T] \right\|_2$, and $b = \max\{b^1, b^2\}$. Then*

$$
b \leq \frac{4}{\min\{n_1, n_2\}}.
$$

*Proof.* We know that $\|e_u\|_2^2 = 1$ and $\|\tilde{e}_i + \tilde{e}_j\|_2^2 = 2$ almost surely. Further,

$$
\mathbb{E}[e_u e_u^T] = \frac{1}{n_1} I_{n_1}, \qquad \mathbb{E}[(\tilde{e}_i + \tilde{e}_j)(\tilde{e}_i + \tilde{e}_j)^T] = \frac{1}{\binom{n_2}{2}}(11^T + (n_2 - 2)I_{n_2})
$$

Using these identities, we get

$$
\begin{aligned}
\mathbb{E}[B^T B] &= \mathbb{E}[(\tilde{e}_i + \tilde{e}_j) e_u^T e_u (\tilde{e}_i + \tilde{e}_j)^T] \\
&= \mathbb{E}_{i,j}[(\tilde{e}_i + \tilde{e}_j) \mathbb{E}_u[e_u^T e_u] w^T] \\
&= \mathbb{E}_{i,j}[(\tilde{e}_i + \tilde{e}_j)(\tilde{e}_i + \tilde{e}_j)^T] \\
&= \frac{1}{\binom{n_2}{2}} (11^T + (n_2 - 2) I_{n_2}) \\
\mathbb{E}[BB^T] &= \mathbb{E}[e_u (\tilde{e}_i + \tilde{e}_j)^T (\tilde{e}_i + \tilde{e}_j) e_u^T] \\
&= \mathbb{E}_u[e_u \mathbb{E}_{i,j}[(\tilde{e}_i + \tilde{e}_j)^T (\tilde{e}_i + \tilde{e}_j)] e_u^T] \\
&= 2\mathbb{E}_u[e_u e_u^T] \\
&= \frac{2}{n_1} I_{n_1}
\end{aligned}
$$

Computing the operator norms of these matrices is straightforward:

$$
b^1 = \left\| \mathbb{E}[B^T B] \right\|_2 = \frac{1}{\binom{n_2}{2}} \left\| 11^T + (n_2 - 2) I_{n_2} \right\|_2 \le \frac{1}{\binom{n_2}{2}} \left( \left\| 11^T \right\|_2 + (n_2 - 2) \left\| I_{n_2} \right\|_2 \right) = \frac{1}{\binom{n_2}{2}} (n_2 + (n_2 - 2)) = \frac{4}{n_2}
$$

$$
b^2 = \left\| \mathbb{E}[BB^T] \right\|_2 = \frac{2}{n_1} \left\| I_{n_1} \right\|_2 = \frac{2}{n_1} \le \frac{4}{n_1}
$$

$$
\therefore b = \max\{b^1, b^2\} \le \frac{4}{\min\{n_1, n_2\}}
$$

$\square$

## D.2. Algebraic Upper Bounds on $\mathcal{D}(WZ^T)$

This subsection contains three lemmas that we shall use in the proof of Lemmas 5.5 and 5.7. The first of these three lemmas, Lemma D.2, gives an upper bound on $\mathcal{D}(WZ^T)$ as a quadratic form around the random matrix $B_{\mathcal{D}}$ that we defined earlier in the section.

Before we state the result, we introduce some additional notation. Corresponding to any matrix $Z \in \mathbb{R}^{n \times r}$, define the vector $z \in \mathbb{R}^n$ as follows:

$$
z_j = \|Z_j\|_2 \ \forall \, j \in [n] \tag{46}
$$

It follows from the definition that

$$
\|z\|_1 = \|Z\|_F^2, \ \|z\|_\infty = \|Z\|_{2,\infty}^2 \tag{47}
$$

Following the convention of splitting the matrix $Z$ into user and item components $Z = (Z_U, Z_V)$, we split the vector $z$ into vectors $z_U \in \mathbb{R}^{n_1}$ and $z_V \in \mathbb{R}^{n_2}$ ($z = (z_U, z_V)$). The norms of these vectors satisfy the following relations:

$$
\|z\|_1 = \|z_U\|_1 + \|z_V\|_1, \ \|z\|_2^2 = \|z_U\|_2^2 + \|z_V\|_2^2, \ \|z\|_\infty = \max\{\|z_U\|_\infty, \|z_V\|_\infty\} \tag{48}
$$

With these notations and identities in place, we proceed to establish the following result.

**Lemma D.2.** *For any two matrices $W$ and $Z$ in $\mathbb{R}^{n \times r}$,*

$$
\mathcal{D}(WZ^T) \le 4(w_U^T B_{\mathcal{D}} z_V + z_U^T B_{\mathcal{D}} w_V)
$$

*Proof.*

$$
\begin{aligned}
\mathcal{D}(WZ^T) &= \frac{1}{m} \sum_{k=1}^{m} \langle\!\langle A_k + A_k^T, WZ^T \rangle\!\rangle^2 \quad \text{(by (7))} \\
&= \frac{1}{m} \sum_{(u;i,j)\in\mathcal{D}} (\langle\!\langle W_u, Z_i - Z_j \rangle\!\rangle + \langle\!\langle Z_u, W_i - W_j \rangle\!\rangle)^2 \quad \text{(by (28))} \\
&\leq \frac{2}{m} \sum_{(u;i,j)\in\mathcal{D}} \langle\!\langle W_u, Z_i - Z_j \rangle\!\rangle^2 + \langle\!\langle Z_u, W_i - W_j \rangle\!\rangle^2 \quad \text{(by } (a+b)^2 \leq 2(a^2+b^2)) \\
&\leq \frac{2}{m} \sum_{(u;i,j)\in\mathcal{D}} \|W_u\|_2^2 \|Z_i - Z_j\|_2^2 + \|Z_u\|_2^2 \|W_i - W_j\|_2^2 \quad \text{(by Cauchy-Schwarz inequality)} \\
&\leq \frac{2}{m} \sum_{(u;i,j)\in\mathcal{D}} \|W_u\|_2^2 (\|Z_i\|_2 + \|Z_j\|_2)^2 + \|Z_u\|_2^2 (\|W_i\|_2 + \|W_j\|_2)^2 \quad \text{(by triangle inequality)} \\
&\leq \frac{4}{m} \sum_{(u;i,j)\in\mathcal{D}} \|W_u\|_2^2 (\|Z_i\|_2^2 + \|Z_j\|_2^2) + \|Z_u\|_2^2 (\|W_i\|_2^2 + \|W_j\|_2^2) \quad \text{(by } (a+b)^2 \leq 2(a^2+b^2)) \\
&= \frac{4}{m} \sum_{(u;i,j)\in\mathcal{D}} w_u(z_i + z_j) + \frac{4}{m} \sum_{(u;i,j)\in\mathcal{D}} z_u(w_i + w_j) \\
&= \frac{4}{m} \sum_{(u;i,j)\in\mathcal{D}} w_U^T \left( e_u(\tilde{e}_i + \tilde{e}_j)^T \right) z_V + \frac{4}{m} \sum_{(u;i,j)\in\mathcal{D}} z_U^T \left( e_u(\tilde{e}_i + \tilde{e}_j)^T \right) w_V \\
&= 4 w_U^T \left( \frac{1}{m} \sum_{(u;i,j)\in\mathcal{D}} e_u(\tilde{e}_i + \tilde{e}_j)^T \right) z_V + 4 z_U^T \left( \frac{1}{m} \sum_{(u;i,j)\in\mathcal{D}} e_u(\tilde{e}_i + \tilde{e}_j)^T \right) w_V \\
&= 4 w_U^T B_\mathcal{D} z_V + 4 z_U^T B_\mathcal{D} w_V
\end{aligned}
$$

$\square$

The next lemma builds upon the previous result to obtain an upper bound in terms of $\left\| B_\mathcal{D} - \bar{B} \right\|_2$.

**Lemma D.3.** *For any* $Z \in \mathbb{R}^{n \times r}$,

$$
\mathcal{D}(ZZ^T) \leq 2 \left( \gamma \|Z\|_F^2 + 2 \left\| B_\mathcal{D} - \bar{B} \right\|_2 \|Z\|_{2,\infty}^2 \right) \|Z\|_F^2
$$

*Proof.* We start by using the relations in (48) along with the arithmetic mean-geometric mean (AM-GM) inequality to obtain the following bound

$$
\|z_U\|_1 \|z_V\|_1 \leq \left( \frac{\|z_U\|_1 + \|z_V\|_1}{2} \right)^2 = \frac{\|z\|_1^2}{4}, \quad \|z_U\|_2 \|z_V\|_2 \leq \left( \frac{\|z_U\|_2 + \|z_V\|_2}{2} \right)^2 \leq \frac{\|z\|_2^2}{2} \tag{49}
$$

Using the bound in (49), we can show the desired result as follows.

$$\frac{\mathcal{D}(ZZ^T)}{8} \leq z_U^T B_{\mathcal{D}} z_V \quad \text{(by Lemma D.2)}$$
$$= z_U^T \bar{B} z_V + z_U^T (B_{\mathcal{D}} - \bar{B}) z_V$$
$$\leq z_U^T \bar{B} z_V + \|z_U\|_2 \left\|(B_{\mathcal{D}} - \bar{B}) z_V\right\|_2 \quad \text{(by the Cauchy-Schwarz inequality)}$$
$$\leq z_U^T \bar{B} z_V + \left\|B_{\mathcal{D}} - \bar{B}\right\|_2 \|z_U\|_2 \|z_V\|_2 \quad \text{(by definition of the operator norm)}$$
$$= \frac{2}{n_1 n_2} z_U^T 11^T z_V + \left\|B_{\mathcal{D}} - \bar{B}\right\|_2 \|z_U\|_2 \|z_V\|_2 \quad \text{(by (44))}$$
$$\leq \gamma z_U^T 11^T z_V + \left\|B_{\mathcal{D}} - \bar{B}\right\|_2 \|z_U\|_2 \|z_V\|_2 \quad (2/(n_1 n_2) \leq 2/(n_1(n_2-1)) = \gamma)$$
$$\leq \gamma \|z_U\|_1 \|z_V\|_1 + \left\|B_{\mathcal{D}} - \bar{B}\right\|_2 \|z_U\|_2 \|z_V\|_2 \quad (1^T z \leq \|z\|_1)$$
$$\leq \frac{1}{4} \left( \gamma \|z\|_1^2 + 2 \left\|B_{\mathcal{D}} - \bar{B}\right\|_2 \|z\|_2^2 \right) \quad \text{(by (49))}$$
$$\leq \frac{1}{4} \left( \gamma \|z\|_1^2 + 2 \left\|B_{\mathcal{D}} - \bar{B}\right\|_2 \|z\|_\infty \|z\|_1 \right) \quad \text{(by Hölder's inequality)}$$
$$= \frac{1}{4} \left( \gamma \|Z\|_F^2 + 2 \left\|B_{\mathcal{D}} - \bar{B}\right\|_2 \|Z\|_{2,\infty}^2 \right) \|Z\|_F^2 \quad \text{(by (47))}$$
$$\therefore \mathcal{D}(ZZ^T) \leq 2 \left( \gamma \|Z\|_F^2 + 2 \left\|B_{\mathcal{D}} - \bar{B}\right\|_2 \|Z\|_{2,\infty}^2 \right) \|Z\|_F^2$$

$\square$

The third and final result of this section builds on Lemma D.2 in a different way as compared to the previous one. Here, we obtain a bound in terms of the $\ell_1$ operator norm of $B_{\mathcal{D}}$. For any matrix $X \in \mathbb{R}^{n_1 \times n_2}$,

$$\|X\|_1 \triangleq \sup_{v:\|v\|_1=1} \|Xv\|_1 \tag{50}$$

It follows that for any $v \in \mathbb{R}^{n_2}$,

$$\|Xv\|_1 \leq \|X\|_1 \|v\|_1 \tag{51}$$

It can be easily shown that

$$\|X\|_1 = \max_{j \in [n_2]} \sum_{i \in [n_1]} |x_{ij}| \tag{52}$$

In addition, we will need Hölder's inequality, which states that for any vectors $a, b$,

$$\langle a, b \rangle \leq \|a\|_\infty \|b\|_1 \Rightarrow \|a\|_2^2 \leq \|a\|_\infty \|a\|_1 \tag{53}$$

Using these inequalities, we get the next result.

**Lemma D.4.** *For any matrices $W, Z \in \mathbb{R}^{n \times r}$,*

$$\mathcal{D}(WZ^T) \leq 4(\max\{\|B_{\mathcal{D}}\|_1, \left\|B_{\mathcal{D}}^T\right\|_1\}) \|Z\|_{2,\infty}^2 \|W\|_F^2,$$

*Proof.* We start by invoking Lemma D.2, we get:

$$\mathcal{D}(WZ^T) \leq 4w_U^T B_{\mathcal{D}} z_V + 4z_U^T B_{\mathcal{D}} w_V$$
$$= 4z_V^T B_{\mathcal{D}}^T w_U + 4z_U^T B_{\mathcal{D}} w_V$$

Applying (48), (51) and (53), we get:

$$z_V^T B_{\mathcal{D}}^T w_U = \langle z_V, B_{\mathcal{D}}^T w_U \rangle \leq \|z_V\|_\infty \left\|B_{\mathcal{D}}^T w_U\right\|_1 \leq \|z_V\|_\infty \left\|B_{\mathcal{D}}^T\right\|_1 \|w_U\|_1 \leq \|z\|_\infty \left\|B_{\mathcal{D}}^T\right\|_1 \|w_U\|_1$$
$$z_U^T B_{\mathcal{D}} w_V = \langle z_U, B_{\mathcal{D}} w_V \rangle \leq \|z_U\|_\infty \left\|B_{\mathcal{D}} w_V\right\|_1 \leq \|z_U\|_\infty \|B_{\mathcal{D}}\|_1 \|w_V\|_1 \leq \|z\|_\infty \|B_{\mathcal{D}}\|_1 \|w_V\|_1$$

Putting the above inequalities together, we get the desired result:

$$
\begin{aligned}
\mathcal{D}(WZ^T) &\leq 4z_V^T B_\mathcal{D}^T w_U + 4z_U^T B_\mathcal{D} w_V \\
&\leq 4\left\|z\right\|_\infty \left\|B_\mathcal{D}^T\right\|_1 \left\|w_U\right\|_1 + 4\left\|z\right\|_\infty \left\|B_\mathcal{D}\right\|_1 \left\|w_V\right\|_1 \\
&\leq 4\left\|z\right\|_\infty \left(\max\{\left\|B_\mathcal{D}\right\|_1, \left\|B_\mathcal{D}^T\right\|_1\}\right)(\left\|w_U\right\|_1 + \left\|w_V\right\|_1) \\
&= 4\left\|z\right\|_\infty \left(\max\{\left\|B_\mathcal{D}\right\|_1, \left\|B_\mathcal{D}^T\right\|_1\}\right)\left\|w\right\|_1 \quad \text{(by(48))} \\
&= 4\left\|Z\right\|_{2,\infty}^2 \left(\max\{\left\|B_\mathcal{D}\right\|_1, \left\|B_\mathcal{D}^T\right\|_1\}\right)\left\|W\right\|_F^2 \quad \text{(by(47))}
\end{aligned}
$$

$\square$

### D.3. Norm Bounds on the Dual Sampling Matrix

First, we provide an upper bound on $\left\|B_\mathcal{D} - \bar{B}\right\|_2$. This result will be used in conjunction with Lemma D.3 to prove Lemma 5.5.

**Lemma D.5.** *Let $\epsilon \in (0,1)$ and $\delta \in (0,1)$ be given. Suppose the number of samples $m$ is at least $(5/\epsilon^2)n\log(n/\delta)$. Then, with probability at least $1-\delta$,*

$$
\left\|B_\mathcal{D} - \bar{B}\right\|_2 \leq \frac{\epsilon}{\min\{n_1, n_2\}}
$$

*Proof.* The matrix Bernstein inequality (Lemma A.4) states that

$$
P(\left\|\bar{B}_m - \bar{B}\right\|_2 \geq t) \leq n\exp\left(-\frac{mt^2/2}{v + 2Lt/3}\right),
$$

where $v = \max\{\left\|\mathbb{E}[BB^T]\right\|_2, \left\|\mathbb{E}[B^T B]\right\|_2\}$ and $L = \sup_B \left\|B\right\|_2$. We have already established that $L = \sqrt{2}$ and $v = 4/(\min\{n_1, n_2\})$ (see Lemma D.1). We would like $\left\|\bar{B}_m - \bar{B}\right\|_2$ to be bounded above by $t = \epsilon/(\min\{n_1, n_2\})$ (for some $\epsilon \in (0,1)$) with probability at least $1-\delta$. Therefore, the number of samples $m$ must satisfy:

$$
\begin{aligned}
n\exp\left(-\frac{mt^2/2}{v + 2Lt/3}\right) &\leq \delta \\
\Leftrightarrow \frac{mt^2/2}{v + 2Lt/3} &\geq \log\left(\frac{n}{\delta}\right)
\end{aligned}
$$

Plugging in the value $L = \sqrt{2}$ and noting that $v = 4t/\epsilon$, we get

$$
\begin{aligned}
\frac{mt^2/2}{(4t/\epsilon) + 2\sqrt{2}t/3} &\geq \log\left(\frac{n}{\delta}\right) \\
\Leftrightarrow \frac{m}{4/\epsilon + 2\sqrt{2}/3} &\geq \frac{2}{t}\log\left(\frac{n}{\delta}\right) \\
\Leftrightarrow m &\geq \left(\frac{4}{\epsilon} + \frac{2\sqrt{2}}{3}\right)\frac{2\min\{n_1, n_2\}}{\epsilon}\log\left(\frac{n}{\delta}\right)
\end{aligned}
$$

Finally, note that $4 + 2\sqrt{2}\epsilon/3 \leq 5$ ($\because \epsilon < 1$) and $2\min\{n_1, n_2\} \leq n_1 + n_2 = n$. Therefore, $m \geq (5/\epsilon^2)n\log(n/\delta)$ is a sufficient condition for the concentration result to hold. $\square$

Next, we move on to proving a high probability bound on $\max\{\left\|B_\mathcal{D}\right\|_1, \left\|B_\mathcal{D}^T\right\|_1\}$. This result will be used in conjunction with Lemma D.4 to prove Lemma 5.7.

For this result, we need to introduce some new notation and some basic inequalities. Define the random matrix $C \in \mathbb{R}^{d_1 \times d_2}$ as follows:

$$
C = \frac{1}{m}\sum_{k=1}^m e_{i_k}\tilde{e}_{j_k}^T \tag{54}
$$

where $(i_k)_{k \in [m]}$ are sampled i.i.d. uniformly at random from $[n_1]$ and $(j_k)_{k \in [m]}$ are sampled i.i.d. uniformly at random from $[n_2]$, independent of $(i_k)_{k \in [m]}$. Let $C_i \in \mathbb{R}^{n_2}$ denote the $i^{\text{th}}$ row of $C$, but expressed as a column vector. Then

$$C_i = \frac{1}{m} \sum_{k=1}^{m} \mathbf{1}_{i_k=i} \tilde{e}_{j_k} \tag{55}$$

It follows that

$$\|C_i\|_1 = \frac{1}{m} \sum_{k=1}^{m} \mathbf{1}_{i_k=i} \tag{56}$$

Note that $\|C_i\|_1$ is the empirical mean of $m$ i.i.d. Bernoulli random variables of mean $1/n_1$. Thus, we can bound it from above by the Chernoff bound.

**Lemma D.6** (Chernoff bound). *Suppose $x_1, x_2, \ldots, x_m$ are i.i.d. Bernoulli random variables with parameter $p$ and let $\epsilon > 0$ be given. Then:*

$$P\left( \frac{1}{m} \sum_{k=1}^{m} x_k \geq p + \epsilon \right) \leq \exp\left( -\frac{m\epsilon^2}{2p(1-p)} \right)$$

Using Lemma D.6 with $p = \epsilon = 1/n_1$, we get that for any $i \in [n_1]$,

$$P\left( \|C_i\|_1 \geq \frac{2}{n_1} \right) \leq \exp\left( -\frac{m}{2n_1} \right)$$

Using the union bound, it follows that

$$P\left( \max_{i \in [n_1]} \|C_i\|_1 \geq \frac{2}{n_1} \right) \leq n_1 \exp\left( -\frac{m}{2n_1} \right)$$

Finally, by (52), we know that

$$\left\| C^T \right\|_1 = \max_{i \in [n_1]} \|C_i\|_1$$

In conclusion,

$$P\left( \left\| C^T \right\|_1 \geq \frac{2}{n_1} \right) \leq n_1 \exp\left( -\frac{m}{2n_1} \right) \tag{57}$$

Since $n_1$ and $n_2$ are arbitrary in the above analysis, one can use the same logic to show that

$$P\left( \|C\|_1 \geq \frac{2}{n_2} \right) \leq n_2 \exp\left( -\frac{m}{2n_2} \right) \tag{58}$$

**Lemma D.7.** *Suppose the number of samples $m$ is at least $2n \log(4n/\delta)$. Then, with probability at least $1 - \delta$,*

$$\max\{\|B_\mathcal{D}\|_1, \|B_\mathcal{D}^T\|_1\} \leq \frac{4}{\min\{n_1, n_2\}}$$

*Proof.* Define the following two matrices

$$B_\mathcal{D}^1 = \frac{1}{m} \sum_{(u;i,j) \in \mathcal{D}} e_u \tilde{e}_i^T ; \quad B_\mathcal{D}^2 = \frac{1}{m} \sum_{(u;i,j) \in \mathcal{D}} e_u \tilde{e}_j^T$$

Both $B_\mathcal{D}^1$ and $B_\mathcal{D}^1$ are statistically identical to the random matrix $C$ defined in (54). By (58), we have that if $m \geq 2n_2 \log(4n_2/\delta)$,

$$P\left( \|B_\mathcal{D}^1\|_1 \geq \frac{2}{n_2} \right) \leq \frac{\delta}{4}, \qquad P\left( \|B_\mathcal{D}^2\|_1 \geq \frac{2}{n_2} \right) \leq \frac{\delta}{4} \tag{59}$$

$$\tag{60}$$

By construction, $B_{\mathcal{D}} = B_{\mathcal{D}}^1 + B_{\mathcal{D}}^2$. By the triangle inequality, we get $\|B_{\mathcal{D}}\|_1 \leq \|B_{\mathcal{D}}^1\|_1 + \|B_{\mathcal{D}}^2\|_1$. Therefore,

$$\|B_{\mathcal{D}}\|_1 \geq \frac{4}{n_2} \Rightarrow \|B_{\mathcal{D}}^1\|_1 + \|B_{\mathcal{D}}^2\|_1 \geq \frac{4}{n_2} \Rightarrow \|B_{\mathcal{D}}^1\|_1 \geq \frac{2}{n_2} \text{ or } \|B_{\mathcal{D}}^2\|_1 \geq \frac{2}{n_2}. \tag{61}$$

Put together, we get that if $m \geq 2n_2 \log(4n_2/\delta)$,

$$P\left(\|B_{\mathcal{D}}\|_1 \geq \frac{4}{n_2}\right) \leq P\left(\|B_{\mathcal{D}}^1\|_1 \geq \frac{2}{n_2} \text{ or } \|B_{\mathcal{D}}^2\|_1 \geq \frac{2}{n_2}\right) \quad \text{(by (61))}$$

$$\leq \mathbb{P}\left(\|B_{\mathcal{D}}^1\|_1 \geq \frac{2}{n_2}\right) + \mathbb{P}\left(\|B_{\mathcal{D}}^2\|_1 \geq \frac{2}{n_2}\right)$$

$$\leq \frac{\delta}{2}. \quad \text{(by (59))}$$

By a similar argument, we can show that if $m \geq 2n_1 \log(4n_1/\delta)$,

$$P\left(\|B_{\mathcal{D}}^T\|_1 \geq \frac{4}{n_1}\right) \leq \frac{\delta}{2}$$

Finally, note that

$$\|B_{\mathcal{D}}\|_1 \leq \frac{4}{n_2} \text{ and } \|B_{\mathcal{D}}^T\|_1 \leq \frac{4}{n_1} \Rightarrow \max\{\|B_{\mathcal{D}}\|_1, \|B_{\mathcal{D}}^T\|_1\} \leq \frac{4}{\min\{n_1, n_2\}}$$

$$\therefore \|B_{\mathcal{D}}^T\|_1 \geq \frac{4}{\min\{n_1, n_2\}} \Rightarrow \|B_{\mathcal{D}}\|_1 \geq \frac{4}{n_2} \text{ or } \|B_{\mathcal{D}}^T\|_1 \geq \frac{4}{n_1}$$

Invoking the union bound once again, we get that if $m \geq 2n \log(4n/\delta)$,

$$P\left(\|B_{\mathcal{D}}^T\|_1 \geq \frac{4}{\min\{n_1, n_2\}}\right) \leq P\left(\|B_{\mathcal{D}}\|_1 \geq \frac{4}{n_2} \text{ or } \|B_{\mathcal{D}}^T\|_1 \geq \frac{4}{n_1}\right)$$

$$\leq P\left(\|B_{\mathcal{D}}\|_1 \geq \frac{4}{n_2}\right) + P\left(\|B_{\mathcal{D}}^T\|_1 \geq \frac{4}{n_1}\right)$$

$$\leq \delta$$

$\square$

### D.4. Proof of Lemma 5.5

**Lemma 5.5.** *Let some $\epsilon, \delta \in (0, 1)$ be given. Suppose the number of samples $m$ exceeds $845 \left(\mu r \kappa/\epsilon\right)^2 n \log\left(n/\delta\right)$. Then, with probability at least $1 - \delta$, $\forall Z \in \overline{\mathcal{C}} \cap \mathcal{B}(\epsilon)$,*

$$\mathcal{D}(\Delta\Delta^T) \leq 10\epsilon\gamma\sigma_r^* \|\Delta\|_F^2.$$

*Proof.* The proof follows from the following facts:

- $\mathcal{D}(\Delta\Delta^T) \leq 2\left(\gamma \|\Delta\|_F^2 + 2\|B_{\mathcal{D}} - \bar{B}\|_2 \|\Delta\|_{2,\infty}^2\right) \|\Delta\|_F^2$, by Lemma D.3.

- $\|\Delta\|_F^2 \leq \epsilon\sigma_r^* \ \forall Z \in \mathcal{B}(\epsilon)$.

- $\|\Delta\|_{2,\infty}^2 \leq 52\mu r \sigma_1^*/n \ \forall Z \in \overline{\mathcal{C}}$. This can be derived as follows.

$$\|\Delta\|_{2,\infty}^2 = \|Z - \Phi\|_{2,\infty}^2 \leq 2\left(\|Z\|_{2,\infty}^2 + \|\Phi\|_{2,\infty}^2\right) \leq 2\left(\frac{12\mu \|Z^*\|_F^2}{n} + \frac{\mu \|Z^*\|_F^2}{n}\right) \leq \frac{52\mu r \sigma_1^*}{n},$$

where the last step follows from the fact that $\|Z^*\|_F^2 \leq 2r\sigma_1^*$.

- The number of samples is at least $5 \left(13\mu r\kappa/\epsilon\right)^2 n \log\left(n/\delta\right)$ $(845 = 5 \cdot 13^2)$. By Lemma D.5, with probability at least $1 - \delta$,

$$\left\|B_{\mathcal{D}} - \bar{B}\right\|_2 \leq \frac{\epsilon}{13\mu r\kappa} \frac{1}{\min\{n_1, n_2\}}$$

Combining these inequalities, we get that with probability at least $1 - \delta$, $\forall\, Z \in \mathcal{B} \cap \bar{\mathcal{C}}$,

$$\mathcal{D}(\Delta\Delta^T) \leq 2\left(\gamma \|\Delta\|_F^2 + 2\left\|B_{\mathcal{D}} - \bar{B}\right\|_2 \|\Delta\|_{2,\infty}^2\right)\|\Delta\|_F^2$$

$$\leq 2\left(\epsilon\gamma\sigma_r^* + 2\frac{\epsilon}{13\mu r\kappa}\frac{1}{\min\{n_1, n_2\}}\frac{52\mu r\sigma_1^*}{n}\right)\|\Delta\|_F^2$$

$$\leq 10\epsilon\gamma\sigma_r^* \|\Delta\|_F^2$$

The last step is reasoned as follows:

$$\frac{2}{n\min\{n_1, n_2\}} = \frac{2}{(n_1 + n_2)\min\{n_1, n_2\}} \leq \frac{2}{\max\{n_1, n_2\}\min\{n_1, n_2\}} = \frac{2}{n_1 n_2} \leq \frac{2}{n_1(n_2 - 1)} = \gamma$$

□

### D.5. Proof of Lemma 5.7

The proof of Lemma 5.7 depends on Lemmas D.4 and D.7.

**Lemma 5.7.** *Suppose the number of samples $m$ is at least $2n\log(4n/\delta)$. Then, with probability at least $1 - \delta$, the following inequalities hold uniformly for all $Z \in \bar{\mathcal{C}}$:*

$$\mathcal{D}(\Delta\Phi^T) \leq 16\gamma(\mu r\sigma_1^*)\|\Delta\|_F^2,$$

$$\mathcal{D}(\Delta\Delta^T) \leq 416\gamma(\mu r\sigma_1^*)\|\Delta\|_F^2,$$

$$\mathcal{D}(WZ^T) \leq 192\gamma(\mu r\sigma_1^*)\|W\|_F^2 \ \forall\, W \in \mathbb{R}^{n\times r}.$$

*Proof.* By Lemma D.4, we have that for any matrices $W, Z \in \mathbb{R}^{n\times r}$,

$$\mathcal{D}(WZ^T) \leq 4(\max\{\|B_{\mathcal{D}}\|_1, \left\|B_{\mathcal{D}}^T\right\|_1\})\|Z\|_{2,\infty}^2\|W\|_F^2,$$

By Lemma D.7 and the assumption on the number of samples we have made, we get that with probability at least $1 - \delta$,

$$\max\{\|B_{\mathcal{D}}\|_1, \left\|B_{\mathcal{D}}^T\right\|_1\} \leq \frac{4}{\min\{n_1, n_2\}}$$

Putting these inequalities together, we get that with probability at least $1 - \delta$, for any matrices $W, Z \in \mathbb{R}^{n\times r}$,

$$\mathcal{D}(WZ^T) \leq \frac{16}{\min\{n_1, n_2\}}\|Z\|_{2,\infty}^2\|W\|_F^2,$$

For the first of the bounds we wish to prove, we replace $W$ by $\Delta$ and $Z$ by $\Phi$. We know that

$$\|\Phi\|_{2,\infty}^2 = \frac{\mu}{n}\|Z^*\|_F^2 \leq \frac{2\mu r\sigma_1^*}{n} \quad (\because \|Z^*\|_F^2 \leq 2r\sigma_1^*)$$

$$\Rightarrow \mathcal{D}(\Delta\Phi^T) \leq \frac{16}{\min\{n_1, n_2\}}\frac{2\mu r\sigma_1^*}{n}\|\Delta\|_F^2 \leq 16\gamma(\mu r\sigma_1^*)\|\Delta\|_F^2$$

Here, as in the proof of Lemma 5.5, we use the fact that $2/(n\min\{n_1, n_2\}) \leq \gamma$. The second and third bounds can be derived in a similar fashion. For the second bound, we use the bound that we established in the proof of Lemma 5.5.

$$\|\Delta\|_{2,\infty}^2 \leq \frac{52\mu r\sigma_1^*}{n}$$

Finally, for the third bound, we use the fact that $Z \in \bar{\mathcal{C}}$ (see Lemma A.3) to get the bound

$$\|\Phi\|_{2,\infty}^2 \leq 12\frac{\mu}{n}\|Z^*\|_F^2 \leq \frac{24\mu r\sigma_1^*}{n}$$

□

## E. Proof of Main Result

In this concluding section, we prove the main results of our paper, namely Lemma 5.1, Lemma 5.2, and Theorem 4.1.

### E.1. Proof of Lemma 5.1

As mentioned in Section 5, Lemma 5.1 follows from Lemmas 5.3, 5.4, and 5.5; these are proven in Appendices B, C, and D respectively. We restate the results here for convenience.

**Lemma 5.3.** *For any $Z \in \overline{\mathcal{C}}$,*

$$\langle\!\langle \nabla \mathcal{L}, \Delta \rangle\!\rangle \geq \frac{\xi}{2} \mathcal{D}\left(\Delta \Phi^T\right) - \frac{5\Xi}{8} \mathcal{D}\left(\Delta \Delta^T\right)$$

**Lemma 5.4.** *Let some $\epsilon, \delta \in (0, 1)$ be given. Suppose the number of samples $m$ exceeds $96\mu r \left(\kappa/\epsilon\right)^2 n \log\left(n/\delta\right)$. Then, with probability at least $1 - \delta$, $\forall\, Z \in \mathcal{H}$,*

$$\mathcal{D}\left(\Delta \Phi^T\right) \geq \gamma \left((1 - \epsilon)\sigma_r^* \|\Delta\|_F^2 + 2\langle\!\langle \Phi_U \Delta_V^T, \Delta_U \Phi_V^T \rangle\!\rangle\right).$$

**Lemma 5.5.** *Let some $\epsilon, \delta \in (0, 1)$ be given. Suppose the number of samples $m$ exceeds $845 \left(\mu r \kappa/\epsilon\right)^2 n \log\left(n/\delta\right)$. Then, with probability at least $1 - \delta$, $\forall\, Z \in \overline{\mathcal{C}} \cap \mathcal{B}(\epsilon)$,*

$$\mathcal{D}(\Delta \Delta^T) \leq 10\epsilon\gamma\sigma_r^* \|\Delta\|_F^2.$$

**Lemma 5.1.** *Suppose the number of samples $m$ is at least $10^7 \left(\mu r \kappa/\tau\right)^2 n \log\left(2n/\delta\right)$, for some $\delta \in (0, 1)$. Then, with probability at least $1 - \delta$, $\forall\, Z \in \mathcal{H} \cap \mathcal{B}(\tau/50) \cap \overline{\mathcal{C}}$,*

$$\langle\!\langle \nabla f, \Delta \rangle\!\rangle \geq \frac{\xi\gamma}{4} \|\Delta\|_F^2 + \frac{\xi\gamma}{8} \left\|\Delta^T D\Phi\right\|_F^2.$$

*Proof.* Our proof strategy will be to put together the statements of the three lemmas and work backwards to calculate the value of the parameter $\epsilon$ needed from Lemmas 5.4 and 5.5 (call them $\epsilon_1$ and $\epsilon_2$ for now). Combining the three aforementioned lemmas gives us:

$$\langle\!\langle \nabla \mathcal{L}, \Delta \rangle\!\rangle \geq \frac{\xi\gamma}{2} \left((1 - \epsilon_1)\sigma_r^* \|\Delta\|_F^2 + 2\langle\!\langle \Phi_U \Delta_V^T, \Delta_U \Phi_V^T \rangle\!\rangle\right) - \frac{25\Xi\gamma}{4} \left(\epsilon_2 \sigma_r^* \|\Delta\|_F^2\right)$$

$$= \frac{2\xi(1 - \epsilon_1) - 25\Xi\epsilon_2}{4} \gamma\sigma_r^* \|\Delta\|_F^2 + \xi\gamma\langle\!\langle \Phi_U \Delta_V^T, \Delta_U \Phi_V^T \rangle\!\rangle$$

Recall from (13) that $\mathcal{R}(Z) = \left\|Z^T DZ\right\|_F^2$. Therefore, $\nabla \mathcal{R}(Z) = 4DZZ^T DZ$. Using this identity and (14), we get:

$$\langle\!\langle \nabla f, \Delta \rangle\!\rangle = \langle\!\langle \nabla \mathcal{L}, \Delta \rangle\!\rangle + \frac{\lambda}{4}\langle\!\langle \nabla \mathcal{R}, \Delta \rangle\!\rangle$$

$$\geq \frac{2\xi(1 - \epsilon_1) - 25\Xi\epsilon_2}{4} \gamma\sigma_r^* \|\Delta\|_F^2 + \xi\gamma\langle\!\langle \Phi_U \Delta_V^T, \Delta_U \Phi_V^T \rangle\!\rangle + \lambda DZZ^T DZ.$$

We focus on the last two terms. Define $\lambda' = \frac{2\lambda}{\xi\gamma}$. Then

$$\xi\gamma\langle\!\langle \Phi_U \Delta_V^T, \Delta_U \Phi_V^T \rangle\!\rangle + \lambda DZZ^T DZ = \frac{\xi\gamma}{2} \left(2\langle\!\langle \Phi_U \Delta_V^T, \Delta_U \Phi_V^T \rangle\!\rangle + \lambda' DZZ^T DZ\right)$$

Following the steps laid out in Zheng & Lafferty (2016) (Appendix C.1), we get the inequality:

$$2\langle\!\langle \Phi_U \Delta_V^T, \Delta_U \Phi_V^T \rangle\!\rangle + \lambda' DZZ^T DZ \geq \frac{\lambda'}{2} \left\|\Phi^T D\Delta\right\|_F^2 - \frac{7\lambda'}{2} \|\Delta\|_F^4 + \left(\lambda' - \frac{1}{2}\right) \mathsf{Tr}(\Phi^T D\Delta\Phi^T D\Delta)$$

We know that $\lambda = \frac{\xi\gamma}{4}$ (see (14)), which implies $\lambda' = 1/2$. Thus, the last term in the above inequality is cancelled out. Plugging this inequality back into the expression above, we get:

$$
\begin{aligned}
\langle\!\langle \nabla f, \Delta \rangle\!\rangle &\geq \frac{2\xi(1-\epsilon_1) - 25\Xi\epsilon_2}{4}\gamma\sigma_r^* \|\Delta\|_F^2 + \xi\gamma\langle\!\langle \Phi_U \Delta_V^T, \Delta_U \Phi_V^T \rangle\!\rangle + \lambda DZZ^T DZ \\
&\geq \frac{2\xi(1-\epsilon_1) - 25\Xi\epsilon_2}{4}\gamma\sigma_r^* \|\Delta\|_F^2 + \frac{\xi\gamma}{2}(2\langle\!\langle \Phi_U \Delta_V^T, \Delta_U \Phi_V^T \rangle\!\rangle + \lambda' DZZ^T DZ) \\
&\geq \frac{2\xi(1-\epsilon_1) - 25\Xi\epsilon_2}{4}\gamma\sigma_r^* \|\Delta\|_F^2 + \frac{\xi\gamma}{2}\left(\frac{1}{4}\left\|\Phi^T D\Delta\right\|_F^2 - \frac{7}{4}\|\Delta\|_F^4\right) \\
&\geq \frac{4\xi(1-\epsilon_1) - 50\Xi\epsilon_2 - 7\xi\epsilon_2}{8}\gamma\sigma_r^* \|\Delta\|_F^2 + \frac{\xi\gamma}{8}\left\|\Phi^T D\Delta\right\|_F^2
\end{aligned}
$$

Choosing $\epsilon_1 = 1/8$ and $\epsilon_2 = \tau/50 = \xi/(50\Xi)$ gives us $4\xi(1-\epsilon_1) - 50\Xi\epsilon_2 - 7\xi\epsilon_2 \geq 2\xi$. Therefore,

$$
\langle\!\langle \nabla f, \Delta \rangle\!\rangle \geq \frac{\xi\gamma}{4}\sigma_r^* \|\Delta\|_F^2 + \frac{\xi\gamma}{8}\left\|\Phi^T D\Delta\right\|_F^2
$$

The number of samples needed for Lemma 5.4 to hold with probability at least $1 - \delta/2$ is

$$
m_1 \geq 96\mu r \left(\kappa/\epsilon_1\right)^2 n \log\left(2n/\delta\right) = 6144\mu r\kappa^2 n \log\left(2n/\delta\right)
$$

The number of samples needed for Lemma 5.5 to hold with probability at least $1 - \delta/2$ is

$$
m_2 \geq 845 \left(\mu r\kappa/\epsilon_2\right)^2 n \log\left(n/\delta\right) \geq 2112500 \left(\mu r\kappa/\tau\right)^2 n \log\left(2n/\delta\right)
$$

The two lemmas jointly hold with probability at least $1 - \delta$. Clearly, the sample complexity requirement from Lemma 5.5 is higher. Thus, we can conclude that given $m \geq 10^7 \left(\mu r\kappa(\Xi/\xi)\right)^2 n \log\left(2n/\delta\right)$ samples, with probability at least $1 - \delta$,

$$
\langle\!\langle \nabla f, \Delta \rangle\!\rangle \geq \frac{\xi\gamma}{4}\sigma_r^* \|\Delta\|_F^2 + \frac{\xi\gamma}{8}\left\|\Phi^T D\Delta\right\|_F^2 \ \forall \ Z \in \mathcal{H} \cup \mathcal{B}(\tau/50) \cup \overline{C},
$$

$\square$

## E.2. Proof of Lemma 5.2

Lemma 5.2 follows from Lemmas 5.6 and 5.7; these are proven in Appendices B and D respectively. We restate the results here for convenience.

**Lemma 5.6.** *For any $Z \in \overline{C}$ and any $W \in \mathbb{R}^{n \times r}$,*

$$
\langle\!\langle \nabla \mathcal{L}, W \rangle\!\rangle^2 \leq 2\Xi^2 \left(\mathcal{D}(\Delta \Phi^T) + \frac{1}{4}\mathcal{D}(\Delta\Delta^T)\right) \mathcal{D}(WZ^T).
$$

**Lemma 5.7.** *Suppose the number of samples $m$ is at least $2n \log(4n/\delta)$. Then, with probability at least $1 - \delta$, the following inequalities hold uniformly for all $Z \in \overline{C}$:*

$$
\begin{aligned}
\mathcal{D}(\Delta \Phi^T) &\leq 16\gamma(\mu r\sigma_1^*) \|\Delta\|_F^2, \\
\mathcal{D}(\Delta\Delta^T) &\leq 416\gamma(\mu r\sigma_1^*) \|\Delta\|_F^2, \\
\mathcal{D}(WZ^T) &\leq 192\gamma(\mu r\sigma_1^*) \|W\|_F^2 \ \forall \ W \in \mathbb{R}^{n \times r}.
\end{aligned}
$$

**Lemma 5.2.** *Suppose the number of samples $m$ is at least $2n \log(4n/\delta)$, for some $\delta \in (0, 1)$. Then, with probability at least $1 - \delta$, $\forall \ Z \in \mathcal{B}(1) \cap \overline{C}$,*

$$
\|\nabla f\|_F^2 \leq 10^5 (\Xi\gamma\mu r\sigma_1^*)^2 \|\Delta\|_F^2 + \frac{(\xi\gamma)^2}{2}\sigma_1^* \left\|\Phi^T D\Delta\right\|_F^2.
$$

*Proof.* From (14), we get that

$$\nabla f = \nabla \mathcal{L} + \frac{\lambda}{4} \nabla \mathcal{R} = \nabla \mathcal{L} + \lambda D Z Z^T D Z$$

$$\therefore \|\nabla f\|_F^2 = \left\| \nabla \mathcal{L} + \lambda D Z Z^T D Z \right\|_F^2 \leq \left( \|\nabla \mathcal{L}\|_F + \left\| \lambda D Z Z^T D Z \right\|_F \right)^2$$

$$\leq 2 \left( \|\nabla \mathcal{L}\|_F^2 + \lambda^2 \left\| D Z Z^T D Z \right\|_F^2 \right) \tag{62}$$

We have assumed that $Z \in \mathcal{B}(1)$, which implies $\|\Delta\|_F^2 \leq \sigma_r^* \leq \sigma_1^*$. Using this bound along with the analysis in Zheng & Lafferty (2016) (Appendix C.2), we get:

$$\left\| D Z Z^T D Z \right\|_F^2 \leq 6 (\|\Delta\|_F^2 + 4\sigma_1^*) \|\Delta\|_F^2 \|Z\|_2^2 + 4\sigma_1^* \left\| \Phi^T D \Delta \right\|_F^2$$

$$\leq 30\sigma_1^* \|\Delta\|_F^2 \|Z\|_2^2 + 4\sigma_1^* \left\| \Phi^T D \Delta \right\|_F^2$$

$$\leq 180 (\sigma_1^*)^2 \|\Delta\|_F^2 + 4\sigma_1^* \left\| \Phi^T D \Delta \right\|_F^2 \quad (\|Z\|_2^2 \leq 6\sigma_1^*) \tag{63}$$

The last bound can be derived as follows:

$$\|Z\|_2^2 = \|\Phi + \Delta\|_2^2 \leq (\|\Phi\|_2 + \|\Delta\|_2)^2 \leq 2(\|\Phi\|_2^2 + \|\Delta\|_2^2) \leq 2(\|\Phi\|_2^2 + \|\Delta\|_F^2) \leq 2(2\sigma_1^* + \sigma_1^*) = 6\sigma_1^*$$

Combining the bounds from Lemma 5.6 and Lemma 5.7, we see that if the number of samples $m$ is at least $2n \log(4n/\delta)$, then with probability at least $1 - \delta$, $\forall Z \in \overline{\mathcal{C}}$,

$$\langle\!\langle \nabla \mathcal{L}, W \rangle\!\rangle^2 \leq 2\Xi^2 \left( \mathcal{D}(\Delta \Phi^T) + \frac{1}{4} \mathcal{D}(\Delta \Delta^T) \right) \mathcal{D}(W Z^T)$$

$$\leq 2\Xi^2 \left( 16\gamma(\mu r \sigma_1^*) \|\Delta\|_F^2 + 104\gamma(\mu r \sigma_1^*) \|\Delta\|_F^2 \right) 192\gamma(\mu r \sigma_1^*) \|W\|_F^2$$

$$= 46080 (\Xi \gamma \mu r \sigma_1^*)^2 \|\Delta\|_F^2 \|W\|_F^2$$

$$\therefore \|\nabla \mathcal{L}\|_F^2 = \sup_{W \in \mathbb{R}^{n \times r}: \|W\|_F = 1} \langle\!\langle \nabla \mathcal{L}, W \rangle\!\rangle^2$$

$$\leq 46080 (\Xi \gamma \mu r \sigma_1^*)^2 \|\Delta\|_F^2 \tag{64}$$

Putting together the bounds in (62), (63), and (64), and plugging in the value of $\lambda = \xi\gamma/4$, we see that if the number of samples $m$ is at least $2n \log(4n/\delta)$, then with probability at least $1 - \delta$, $\forall Z \in \mathcal{B} \cap \overline{\mathcal{C}}$,

$$\|\nabla f\|_F^2 \leq 2 \left( 46080 (\Xi \gamma \mu r \sigma_1^*)^2 \|\Delta\|_F^2 + 12(\xi \gamma \sigma_1^*)^2 \|\Delta\|_F^2 \right) + \frac{(\xi\gamma)^2}{2} \sigma_1^* \left\| \Phi^T D \Delta \right\|_F^2$$

$$\leq 10^5 (\Xi \gamma \mu r \sigma_1^*)^2 \|\Delta\|_F^2 + \frac{(\xi\gamma)^2}{2} \sigma_1^* \left\| \Phi^T D \Delta \right\|_F^2$$

$\square$

### E.3. Proof of Theorem 4.1

Lemmas 5.1 and 5.2 are the two key ingredients needed to prove the main theorem of this paper.

**Theorem 4.1.** *Suppose the following conditions hold:*

- *The dataset $\mathcal{D}$ consists of $m$ i.i.d. samples generated according to the model presented in Section 3.1.*

- *The number of samples $m$ is at least $10^7 (\mu r \kappa/\tau)^2 n \log(8n/\delta)$ for some $\delta \in (0, 1)$.*

- *The initial point $Z^0$ lies in $\mathcal{B}(\tau/50)$.*

- *The stepsize $\eta$ in Algorithm 1 satisfies $\eta\alpha \leq 2.5 \cdot 10^{-6} (\tau/\mu r \kappa)^2$.*

*Then, with probability at least $1 - \delta$, the iterates $Z^1, Z^2, \ldots$ of Algorithm 1 satisfy:*

$$\left\|\Delta(Z^t)\right\|_F^2 \leq \left(1 - \frac{\alpha\eta}{4}\right)^t \left\|\Delta(Z^0)\right\|_F^2 \quad \forall\, t \in \mathbb{N}.$$

*Proof.* We begin by following the standard steps in the analysis of gradient descent.

$$
\begin{aligned}
\left\|\Delta(Z^{t+1})\right\|_F^2 &= \left\|Z^{t+1} - \Phi(Z^{t+1})\right\|_F^2 \\
&\leq \left\|Z^{t+1} - \Phi(Z^t)\right\|_F^2 \\
&= \left\|\mathcal{P}_{\mathcal{H}}(\mathcal{P}_{\mathcal{C}}\left(Z^t - \eta\nabla f(Z^t)\right)) - \Phi(Z^t)\right\|_F^2 \\
&\leq \left\|Z^t - \eta\nabla f(Z^t) - \Phi(Z^t)\right\|_F^2 \\
&= \left\|\Delta(Z^t) - \eta\nabla f(Z^t)\right\|_F^2 \\
&= \left\|\Delta(Z^t)\right\|_F^2 + \eta^2 \left\|\nabla f(Z^t)\right\|_F^2 - 2\eta\langle\!\langle\nabla f(Z^t), \Delta(Z^t)\rangle\!\rangle
\end{aligned}
$$

The first inequality comes from the fact that $\Phi(Z^{t+1})$ is the closest point in $\Phi$ to $Z^{t+1}$; this is by definition of $\Phi(Z^{t+1})$. The second inequality follows from the fact that $\Phi(Z^t) \in \mathcal{C}$ (by Lemma A.2) and $\Phi(Z^t) \in \mathcal{H}$ (by assumption); thus, successive projections of the iterate on to $\mathcal{C}$ and $\mathcal{H}$ can only bring it closer to $\Phi(Z^t)$.

Next, suppose the following bounds hold for some positive constants $a, b, c,$ and $d$ and for all $t \in \mathbb{Z}_+$:

$$\langle\!\langle\nabla f(Z^t), \Delta(Z^t)\rangle\!\rangle \geq a \left\|\Delta(Z^t)\right\|_F^2 + c \left\|\Delta(Z^t)^T D\Phi(Z^t)\right\|_F^2 \tag{65}$$

$$\left\|\nabla f(Z^t)\right\|_F^2 \leq b \left\|\Delta(Z^t)\right\|_F^2 + d \left\|\Delta(Z^t)^T D\Phi(Z^t)\right\|_F^2 \tag{66}$$

It follows that:

$$
\begin{aligned}
\left\|\Delta(Z^{t+1})\right\|_F^2 &\leq \left\|\Delta(Z^t)\right\|_F^2 + \eta^2 \left\|\nabla f(Z^t)\right\|_F^2 - 2\eta\langle\!\langle\nabla f(Z^t), \Delta(Z^t)\rangle\!\rangle \\
&\leq (1 - 2\eta a + \eta^2 b) \left\|\Delta(Z^t)\right\|_F^2 + (\eta^2 d - 2\eta c) \left\|\Delta(Z^t)^T D\Phi(Z^t)\right\|_F^2 \\
&\leq (1 - \eta a) \left\|\Delta(Z^t)\right\|_F^2,
\end{aligned}
\tag{67}
$$

provided $\eta \leq \min(a/b, 2c/d)$. The last step can be justified as follows:

$$\eta \leq \frac{a}{b} \Rightarrow (1 - 2\eta a + \eta^2 b) \leq 1 - \eta a, \quad \eta \leq \frac{2c}{d} \Rightarrow \eta^2 d - 2\eta c \leq 0$$

Further, if $\eta \leq 1/a$, then $1 - \eta a \geq 0$, implying that the right-hand side of (67) remains positive. This allows us to use the inequality repeatedly to yield:

$$\left\|\Delta(Z^t)\right\|_F^2 \leq (1 - \eta a)^t \left\|\Delta(Z^0)\right\|_F^2 \quad \forall\, t \in \mathbb{Z}_+$$

Finally, observe that we have assumed the number of samples given, $m$, is at least $10^7 \left(\mu r \kappa / \tau\right)^2 n \log\left(8n/\delta\right)$. This ensures that with probability at least $1 - \delta$, both Lemmas 5.1 and 5.2 hold. Lemmas 5.1 and 5.2 imply that the inequalities (65) and (66) hold for all $Z \in \mathcal{H} \cap \mathcal{B}(\tau/50) \cap \overline{\mathcal{C}}$ with parameters:

$$a = \frac{\xi\gamma}{4}\sigma_r^*, \quad b = 10^5 (\Xi\gamma\mu r\sigma_1^*)^2, \quad c = \frac{\xi\gamma}{8}, \quad d = \frac{(\xi\gamma)^2}{2}\sigma_1^*$$

Given these parameters, as long as the stepsize $\eta$ satisfies $\eta \leq a/b = 2.5 \cdot 10^{-6}(\tau/\mu r\kappa)^2/\alpha$, the other conditions on $\eta$ are automatically satisfied. $\qquad\square$

## F. Extra Simulation Results

In this section, we present simulation results that highlight the dependency of the sample complexity of the learning problem on the rank $r$ of the ground-truth matrix $X^*$. In this experiment, the parameters used are as follows. The

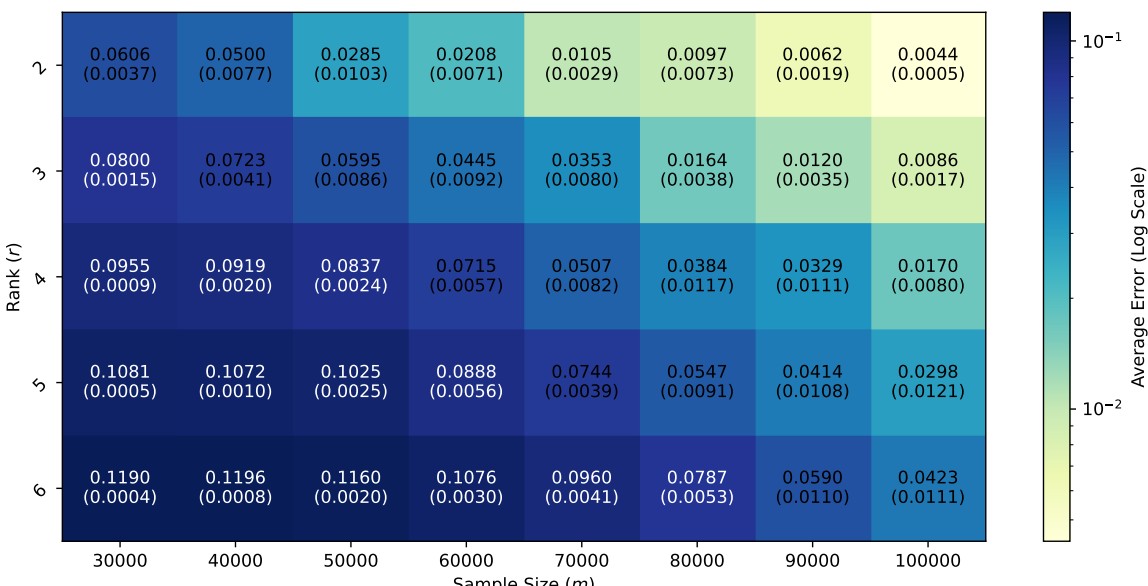

*Figure 2.* Variation of the reconstruction error as a function of the underlying rank of the matrix $r$ and the sample size $m$, for ground-truth matrices of size $(2000, 3000)$.

generated matrices have size $(n_1, n_2) = (2000, 3000)$. We vary the rank $r$ among the values $\{2, 3, \ldots, 6\}$. For every possible value of $r$, we generate a dataset the manner described in Section 3.1, whose size $m$ varies among the values $\{30,000, 40,000, \ldots, 100,000\}$. The comparisons are noiseless. The regularizer coefficient $(\lambda/4)$ was set to $0.01$.

For faster implementation, we optimize the loss function using the PyTorch implementation of Adam instead of using gradient descent. As our experiments in Section 6 show, we need neither a smart initialization nor the projection step. At the end of 300 epochs, we compute the reconstruction error $\|X^t - X^*\|_F / \sqrt{n_1 n_2}$. For each value of $r$ and $m$, we run this experiment with ten fresh seeds. The values reported in Figure 2 are the mean and standard deviation of the reconstruction error across these ten runs.

We observe that the reconstruction error uniformly increases with the rank $r$ and decreases with the sample size $m$. Interestingly, if we observe the boxes with roughly the same error, we see that the sample complexity increases roughly linearly with the rank $r$. This is not surprising, as ultimately the matrix to be estimated is $Z^*$, which has $nr$ parameters. However, our theoretical analysis gives us a sample complexity that grows as $O(nr^2)$. Thus, there is scope for further research in order to develop $O(nr)$ sample complexity guarantees.

