# OpenReview forum: "Recommendations with Sparse Comparison Data: Provably Fast Convergence for Nonconvex Matrix Factorization"
_ICML.cc/2025/Conference — ICML 2025 poster_

### Official Review · Reviewer_Rcd3 · 2025-03-10

**Overall Recommendation:** 3

**Summary:**

This paper addresses comparison-based recommendations using non-convex matrix factorization optimization. While this approach is more efficient than convex optimization, it remains challenging. The authors observe that although finding the global minimum may be difficult, the non-convex function behaves convexly near the true solution

**Claims And Evidence:**

This work begins optimization with a noiseless warm start, which seems unlikely to be practical in industrial settings or real-world datasets. Can the authors also compare results using naive gradient descent versus the projected method?

**Essential References Not Discussed:**

Although the paper references prior work that informs its approach, it lacks a comparison with other baseline methods to evaluate its results.

**Experimental Designs Or Analyses:**

Figure 1 summarizes the simulation results, which align well with the theoretical hypothesis and proofs.

**Methods And Evaluation Criteria:**

The authors test their approach by simulating a random ground truth matrix drawn from a normal distribution, justified by the presence of popularity bias in user preferences. Figure 1 shows that convergence becomes more linear for large datasets.

**Other Comments Or Suggestions:**

no

**Other Strengths And Weaknesses:**

1. The paper is well-written and easy to follow.
2. It provides solid theoretical proofs.
3. The paper's first assumption is that the rating matrix is noiseless. Given that most real-world recommendation datasets are noisy, or at least somewhat, this assumption may not be realistic, particularly in industry settings.
4. The results in Figure 1 do not account for other hyperparameters, such as the projection step.

**Questions For Authors:**

Most of the theoretical proofs assume a non-noisy setting. I wonder how introducing a hyperparameter to adjust for noise in the ground truth matrix would affect the results in Figure 1.

**Relation To Broader Scientific Literature:**

This work proposes a probabilistic model to address comparison-based recommendation using a non-convex loss. One key assumption is that the rating matrix is noise-free or that optimization begins with a warm start. While this assumption may limit real-world applicability, the work can serve as a solid baseline for future studies to explore probabilistic approaches with cold starts and noisy data, which are more common in both industry and academia. Additionally, given the ground-truth matrix's structure—representing user preferences between two items—future research could extend the model to support k-item comparisons.

**Theoretical Claims:**

I reviewed the theoretical proofs at the board level. The authors present a clear and logical flow to justify their approach.

---

> ### Author Rebuttal · Authors · 2025-03-31
>
> Thank you for your careful review and appreciation of our writing and theoretical analysis. We understand your concerns regarding the warm start and noiseless assumptions and address them below. We also address your questions about with the projection step.
>
> Before addressing your concerns, we restate our main result: in a neighborhood of the ground-truth solution, the loss function is strongly convex. Thus, when initialized within this neighborhood, projected gradient descent converges linearly to the ground truth. This is the first such result for the learning from comparisons problem. In fact, the main takeaway of our paper is a very practical message: it is possible to learn personalized user preferences from comparison data in a computationally efficient and statistically efficient manner. As you noted, our work serves as a baseline for future theoretical studies to establish guarantees with cold starts and noisy data. We hope this will be the case and are actively working toward such generalizations, though these extensions are nontrivial. The challenges can be seen in the matrix completion literature, where initial assumptions, similar to ours, were later relaxed by follow-up work (see Section 1.1 of our paper). However, broadly speaking, these assumptions are largely for analytical convenience; they are not needed in practice. Our simulations aim to highlight this fact.
>
> Regarding the projection step: we need it to ensure that iterates remain in the set of incoherent matrices; this, in turn, is necessary for our theoretical analysis. If iterates remain incoherent naturally via gradient descent, projection is unnecessary. Our simulations (Figures 1a, 1c) confirm that vanilla gradient descent also converges to the true solution. The strong overlap of error curves for projected and vanilla gradient descent indicates that the projection step hardly perturbs the iterates. In matrix completion, early theoretical works assumed projection steps or regularization to maintain incoherence, but later research showed these were not necessary. Ma et al. (2020) demonstrated that gradient descent has implicit regularization, leading to convergence without explicit projection. Extending this implicit regularization result to learning from comparisons is a promising research direction.
>
> The warm start assumption ensures initialization in a strongly convex region, allowing us to prove linear convergence of gradient descent. However, our experiments (Figures 1a, 1c) indicate that this assumption is not needed in practice, as even random initialization leads to linear convergence. This phenomenon has also been observed in matrix completion. Early work (2010–2015) on nonconvex matrix factorization assumed a warm start to establish theoretical guarantees, which was later relaxed by Ge et al. (2016–2017), who showed that all local minima are global. Since gradient descent finds local minima, this explains why warm starts are not necessary in practice. However, their analysis relies on quadratic loss functions, making extensions to our setting nontrivial. Furthermore, global analyses of matrix completion do not provide convergence rates, whereas our work guarantees linear convergence within a specific region.
>
> The noiseless assumption is primarily for analytical convenience; our algorithm can be applied directly to noisy data without modification. This assumption enables us to show exact convergence to the ground truth. Without it, convergence can only be guaranteed up to some statistical estimation error. Analyzing the noisy setting is an important but separate research problem that will likely build on the key ideas introduced in our work. While our original submission lacked simulations on noisy data, we have since conducted extensive experiments showing that performance degrades gracefully with noise (noise in the scores and noisy comparisons). Specifically, when adding a small noise matrix to the ground-truth low-rank matrix, gradient descent converges to a point where the residual error is nearly equal to the norm of the noise matrix. This suggests our method effectively learns the best low-rank approximation of the noisy ground truth. We will include these results in the revised paper. (Unfortunately, we are not able to convey meaningful results through markdown tables here.)
>
> In summary, we assume a warm start, noiseless observations, and the projection step to simplify the analysis. However, our experiments suggest these assumptions are not necessary in practice. Even without them, simple gradient descent converges efficiently, making learning from comparisons a practical and robust approach. We hope this rebuttal (along with the other rebuttals) addresses the major concerns you have of our paper. We shall be happy to answer any other questions you may have.

---

### Official Review · Reviewer_aNun · 2025-03-12

**Overall Recommendation:** 3

**Summary:**

This paper focuses on the nonconvex learning problem in recommendation systems based on pairwise user comparison feedback, which has often been formulated as a convex optimization over utility matrices in prior literature. The authors propose a nonconvex matrix factorization approach to model pairwise comparisons as noisy evaluations based on the difference in latent utilities and to solve a maximum likelihood formulation over the latent factors using a nonconvex optimization approach. The main theoretical contribution is a proof that, under a warm start and in a sparse data regime, the negative log-likelihood objective exhibits a locally strong convexity–like behavior. This property guarantees that a projected gradient descent method converges exponentially fast to a solution equivalent to the ground truth. The paper provides detailed proofs employing matrix concentration inequalities (in particular, a matrix Bernstein inequality) and supports the theoretical findings with simulation results on synthetic data.

## update after rebuttal
I will keep the scores unchanged.

**Claims And Evidence:**

* **Claims:** The key claim of this papaer is that the nonconvex formulation for learning from sparse comparison data exhibits (restricted) strong convexity in a neighborhood of the true solution, which guarantees exponential convergence of projected gradient descent given a warm start. Additional claims include sample complexity bounds and explicit dependence on parameters such as the incoherence and condition number.

* **Evidence:** The authors back these claims with detailed theoretical derivations (Theorem 3.1, supporting Lemmas 4.1–4.7 and their proofs in the appendix) accompanied by simulation experiments that validate the exponential convergence rate.

All the claims are well motivated and supported by rigorous proofs.

**Essential References Not Discussed:**

Based on my knowledge, the related work discussed in this paper has comprehensive coverage and adequately demonstrates the paper's contributions.

**Experimental Designs Or Analyses:**

The experiments use synthetic data in both low- and high-dimensional settings, simulating pairwise comparisons using a known ground-truth matrix. The simulation results confirm that, when using the recommended step-size and sample complexity, the algorithm exhibits linear (exponential) convergence as predicted by the theory.

While the synthetic experiments provide compelling evidence for the theoretical findings, given that recommender systems are highly practical applications and considering that most theoretical works mentioned in this paper have been tested on real-world datasets, the absence of experiments on real-world comparison data is a potential concern.

**Methods And Evaluation Criteria:**

* **Methods:** The authors propose a projected gradient descent method with two projection steps: one to enforce incoherence of the iterates and one to remove the shift invariance (due to comparing only differences).
* **Evaluation Criteria:** The theoretical results are evaluated using concentration inequalities and error bounds derived via the matrix Bernstein inequality. The experimental evaluation measures convergence speed via normalized Frobenius norm errors over iterations.

**Other Comments Or Suggestions:**

It is recommended that the authors demonstrate empirical validation on real-world datasets. If existing datasets do not meet the assumptions of this paper, or if there are other reasons that only synthetic experiments can be conducted, please explicitly explain these limitations.

**Other Strengths And Weaknesses:**

The theoretical guarantees provided in this paper for the nonconvex formulation—especially in the sparse data regime—are a clear and noteworthy contribution to studies on pairwise comparisons in recommendation systems. However, while the theoretical aspects are strong, the lack of empirical validation on real-world comparison datasets somewhat limits the practical implications of the work, particularly considering the applied nature of recommender systems.

**Questions For Authors:**

N/A

**Relation To Broader Scientific Literature:**

The paper is situated at the intersection of nonconvex matrix factorization and learning from comparison data. It represents an important extension of [1], addressing the scalability issues of previous works [2,3] on large-scale datasets.

[1] Negahban, S., Oh, S., Thekumparampil, K. K., and Xu,J. Learning from comparisons and choices. Journal of Machine Learning Research, 19(40):1–95, 2018.

[2] Park, D., Neeman, J., Zhang, J., Sanghavi, S., and Dhillon, I. Preference completion: Large-scale collaborative ranking from pairwise comparisons. In Proceedings of the 32nd International Conference on Machine Learning, 2015.

[3] Rendle, S., Freudenthaler, C., Gantner, Z., and Schmidt Thieme, L. BPR: Bayesian personalized ranking from implicit feedback. In Proceedings of the Twenty-Fifth Conference on Uncertainty in Artificial Intelligence, 2009.

**Theoretical Claims:**

I did my best to review the derivation of key intermediate results (particularly Lemma 4.3, 4.4, 4.5) under the stated assumptions, and found the arguments to be mathematically sound given the stated assumptions.

---

> ### Author Rebuttal · Authors · 2025-03-31
>
> We sincerely appreciate your time and effort in reviewing our paper and recognizing its theoretical contributions. We also acknowledge your suggestion that experiments on real-world datasets would further strengthen our work. While we generally agree, we have chosen not to include such experiments here for the following reasons.
>
> First, the primary focus of this paper is on analyzing the nonconvex optimization problem that arises in the context of learning from comparisons. The core challenge is to establish that the loss landscape has favorable structural properties, ensuring gradient descent quickly converges to the global minimum. To focus on this aspect, we assume a low-rank matrix model for utilities and an oracle choice model, both of which are widely accepted in the recommender systems literature. While real-data experiments could complement our results, they would also introduce confounding factors such as how well the assumed model fits reality and the extent of noise in the data. To avoid this confusion, we work exclusively with synthetic data, where these factors do not crop up. Through our experiments, we demonstrate that some of the assumptions needed for the theoretical analysis (warm start, projection step) are not necessary in practice, which improves the practical viability of this method. Our work demonstrates that learning personalized preferences from comparison data is both statistically efficient (low sample complexity) and computationally efficient (exponentially fast gradient descent).
>
> Second, we are not the first to study the problem of learning personalized rankings from comparison data; both Rendle et al. (2009) and Park et al. (2015) study this very problem. They use the same factorized optimization approach as our work and test it on real-world datasets. In terms of the method, therefore, there is no substantial difference between this prior work and ours. However, before our work, these methods were essentially heuristics. We provide a solid theoretical foundation for why gradient descent on factorized matrices leads to good solutions, a nontrivial result given the nonconvexity of the problem. Despite extensive prior work on nonconvex optimization—especially in matrix completion—existing results do not apply to the learning-from-comparisons setting. Our work addresses this open problem. It is also useful to draw a parallel with the research on classical matrix completion: empirical success of matrix factorization for recommendations (Mnih & Salakhutdinov, 2007; Koren et al., 2009) preceded theoretical justification, which emerged gradually through successive refinements.
>
> Finally, an important limitation in the field is the lack of explicit comparison datasets. To the best of our knowledge, there is no dataset in the regime of recommender systems where users are \textit{explicitly} asked to compare two items according to their preference. Note that both Rendle et al. (2009) and Park et al. (2015) \textit{infer} comparisons from other forms of data. In Rendle et al. (2009), items that a user has viewed/purchased are interpreted to be more preferred than those that the user has not viewed. In Park et al. (2015), a user is said to have preferred one item over another if it rates the first item higher than the second. While these inferred comparisons have their merits, our work suggests that explicitly collecting comparison data could also be an effective approach; it could potentially be more effective learning from ratings. Curating such a dataset and testing this hypothesis is an important direction for future work. In the absence of such datasets, however, we would have to resort to inferring comparisons from ratings/views, as done by prior work. Doing so would merely amount to reproducing the results of Rendle et al. (2009) and Park et al. (2015), given the large similarity of the methods. Thus, we have not performed such experiments here.
>
> We hope this rebuttal helps you better appreciate the contribution of our paper. We shall be happy to answer any other questions you may have.

---

### Official Review · Reviewer_9TXS · 2025-03-14

**Overall Recommendation:** 4

**Summary:**

In practical settings, users are often picking between their favorite of a few items. As such, we learn about a user’s preferences via the comparisons they made. Given features about the users and the items, the objective is to recover the low-rank matrix of information given data points of the format (user, (item 1, item 2), favorite expected outcome). Specifically in this setting, the authors assume that the outcomes are noiseless, meaning that in fact the expected outcome is revealed. Another assumption made is that we are given an initial matrix which is in an epsilon-ball in the Frobenius norm around the ground-truth matrix. The authors provide theoretical analysis of the non-convex formulation of this problem and provide an algorithm to solve the problem under the assumptions made. The authors also empirically validate their algorithm.

**Claims And Evidence:**

Yes, the claims are supported by clear and convincing evidence.

**Essential References Not Discussed:**

n/a

**Experimental Designs Or Analyses:**

The experimental design for the most part is sound and valid. However, it would be enlightening also to include different values for $r$, the rank of the underlying matrix.

**Methods And Evaluation Criteria:**

Yes, the proposed methods make sense. Specifically, the algorithm is supported by carefully written theorems.

**Other Comments Or Suggestions:**

See above

**Other Strengths And Weaknesses:**

Strengths:
* The paper addresses an open problem, which is the matrix factorization from comparison data in the non-convex setting. Although a few simplifying assumptions are made in order to achieve this result, these are still meaningful steps in solving the more general problem. The algorithm’s dependence on the problem size is $O(nr^2 \log n)$ where $n$ is the number of samples and $r$ is the rank of the matrix, and converges exponentially fast. The algorithm itself makes two projections in order to stay in the correct region of matrices, and shows that these projections maintain the correct invariance throughout the algorithm.
* The analysis is overall clear and easy to follow.

Weaknesses:
* Although the paper is for the most part clear, some improvements could be made in Sections 2.1 and 2.2 to properly motivate the problem setup. In particular, why should $Z^*$ be considered the ground truth matrix, rather than $X^*$? Also, it would be nice to add some motivation for the following definitions such as $Y^*$ as well, and what role these matrices play later on. Furthermore, it would be helpful to the reader to state exactly the definition of $A_k$.

**Questions For Authors:**

See above

**Relation To Broader Scientific Literature:**

This paper addresses an open problem and provides an algorithm to solve it under some assumptions.

**Theoretical Claims:**

I have checked the correctness of the claims made in the paper.

---

> ### Author Rebuttal · Authors · 2025-03-31
>
> We thank you for a thorough and positive review of our paper. Here, we address the couple of concerns you have raised. First, you mentioned "it would be enlightening also to include different values for $r$, the rank of the underlying matrix." We agree. Below, we give a table highlighting the estimation error as a function of the rank and the number of samples. We observe that to get the same estimation error, the number of samples grows (approximately linearly) with the rank. In the revised version of the paper, we shall include this experimental result in the form of a heatmap, where this relationship is easy to observe. Second, you mentioned "some improvements could be made in Sections 2.1 and 2.2 to properly motivate the problem setup." Thank you for this feedback, we shall improve these sections in the final version. Indeed, in the current form, some of the definitions may seem a bit arbitrary without further justification.
>
> To answer your specific questions: "Why should $Z^*$ be considered the ground truth matrix, rather than $X^*$? What is the motivation behind defining $Y^*$?" Both these questions are related. The analysis crucially relies on analyzing symmetric matrices $Y$, which can be factored as $Y = ZZ^T$. To elaborate: we work with terms of the form $\mathcal{D}(Y)$ (defined in (21)), and prove our concentration results for such terms; for these results, $Y$ must be a symmetric matrix. Fundamentally, however, we are trying to estimate the asymmetric matrix $X^* = U^*{V^*}^T$. To overcome this gap, we transform the problem of estimating $X^*$ to one of estimating $Y^*$ (a bigger, but equivalent matrix), which in turn is reduced to the problem of estimating $Z^*$ (because $Y^* = Z^* Z^{*T}$). Thus, ultimately, our loss function is in terms of $Z$ (see (12)). We briefly allude to the relation between these matrices in the first paragraph of Section 1.3, but we will reiterate these connections again in Section 2.1 of our revision. This particular transformation is first proposed in the paper of Zheng and Lafferty, and we borrow this idea from there. Finally, $A_1, A_2, \ldots, A_m$ are i.i.d. random matrices, all of the same form as $A$ (described in equation (8)).
>
> We now present a table highlighting the performance of our algorithm as a function of the underlying rank $r$ and the number of samples $m$. We generated a matrix of size 2000x3000 in the same manner as described in Section 5 of the paper. The numbers shown below are the estimation errors, averaged over ten independent runs.
>
> | Rank ↓ / Samples → | 30000         | 40000         | 50000         | 60000         | 70000         | 80000         | 90000         | 100000        |
> |--------------------|--------------|--------------|--------------|--------------|--------------|--------------|--------------|--------------|
> | 2  | 0.0606 ± 0.0037 | 0.0500 ± 0.0077 | 0.0285 ± 0.0103 | 0.0208 ± 0.0071 | 0.0105 ± 0.0029 | 0.0097 ± 0.0073 | 0.0062 ± 0.0019 | 0.0044 ± 0.0005 |
> | 3  | 0.0800 ± 0.0015 | 0.0723 ± 0.0041 | 0.0595 ± 0.0086 | 0.0445 ± 0.0092 | 0.0353 ± 0.0080 | 0.0164 ± 0.0038 | 0.0120 ± 0.0035 | 0.0086 ± 0.0017 |
> | 4  | 0.0955 ± 0.0009 | 0.0919 ± 0.0020 | 0.0837 ± 0.0024 | 0.0715 ± 0.0057 | 0.0507 ± 0.0082 | 0.0384 ± 0.0117 | 0.0329 ± 0.0111 | 0.0170 ± 0.0080 |
> | 5  | 0.1081 ± 0.0005 | 0.1072 ± 0.0010 | 0.1025 ± 0.0025 | 0.0888 ± 0.0056 | 0.0744 ± 0.0039 | 0.0547 ± 0.0091 | 0.0414 ± 0.0108 | 0.0298 ± 0.0121 |
> | 6  | 0.1190 ± 0.0004 | 0.1196 ± 0.0008 | 0.1160 ± 0.0020 | 0.1076 ± 0.0030 | 0.0960 ± 0.0041 | 0.0787 ± 0.0053 | 0.0590 ± 0.0110 | 0.0423 ± 0.0111 |

---

### Decision · Program_Chairs · 2025-05-01

**Decision:**

Accept (poster)

**Comment:**

This paper focuses on the nonconvex learning problem in recommendation systems based on pairwise user comparison feedback, which has often been formulated as a convex optimization over utility matrices in prior literature. The authors propose a nonconvex matrix factorization approach to model pairwise comparisons as noisy evaluations based on the difference in latent utilities and to solve a maximum likelihood formulation over the latent factors using a nonconvex optimization approach. The main theoretical contribution is a proof that, under a warm start and in a sparse data regime, the negative log-likelihood objective exhibits a locally strong convexity–like behavior. This property guarantees that a projected gradient descent method converges exponentially fast to a solution equivalent to the ground truth. The paper provides detailed proofs employing matrix concentration inequalities (in particular, a matrix Bernstein inequality) and supports the theoretical findings with simulation results on synthetic data.


Strengths are: 1) The paper addresses an open problem, which is the matrix factorization from comparison data in the non-convex setting; 2) The theoretical guarantees provided in this paper for the nonconvex formulation—especially in the sparse data regime—are a clear and noteworthy contribution to studies on pairwise comparisons in recommendation systems; 3) The analysis is overall clear and easy to follow; 4) The paper is well-written and easy to follow. One major concern is that the paper's first assumption is that the rating matrix is noiseless. Given that most real-world recommendation datasets are noisy, or at least somewhat, this assumption may not be realistic, particularly in industry settings. Overall, although a few simplifying assumptions are made in order to achieve this result, these are still meaningful steps in solving the more general problem.